# SELF-SUPERVISED LEARNING WITH SPATIALITY PRESERVING REPRESENTATION FOR EEG SIGNALS

## ABSTRACT

Self-supervised learning (SSL) has revolutionized the field of deep learning with EEG signals, yet current approaches face a critical limitation: the loss of crucial spatial information due to architectures that fail to adequately preserve the one-to-one electrode relationship between the input and representation. To address this, we introduce Spatiality Preserving Representation (SPR) Learning. Unlike existing methods relying on reconstruction or temporal prediction with separate encoders, SPR learns spatial relationships through an innovative coherence pseudolabel prediction task, teaching models to understand the intricate topographical organization of brain signals that conventional approaches overlook. Through comprehensive evaluation, SPR demonstrates superior performance over state-of-the-art methods (4.7%, 9.7%, 1.6%, and 15.6% fine-tuning improvements over different datasets), learning meaningful spatial representations that capture the complex spatial-temporal dynamics inherent in EEG data. Our work opens new avenues for interpreting the relationships of different brain regions by prioritizing spatial awareness, and thus bridge the gap between functional connectivity analysis and self-supervised EEG representation learning.

## 1 INTRODUCTION

Electroencephalography (EEG) is a noninvasive neuroimaging technique that captures brain activity by recording electrical signals through electrodes placed on the scalp (Teplan et al., 2002). Compared to other neuroimaging modalities such as functional magnetic resonance imaging (fMRI) and magnetic resonance imaging (MRI), EEG offers greater affordability and accessibility for early disease detection (Bosl et al., 2018; Parmar & Paunwala, 2023; Ehteshamzad, 2024) and serves as particularly valuable diagnostic tools (Jeong, 2004; Britton et al., 2016). Consequently, machine learning (ML) approaches applied to EEG analysis have gained significant research attention in recent years (Rafiei et al., 2022; Hosseini et al., 2021).

Self-supervised learning (SSL) is an ML paradigm that leverages unlabeled data to extract inherent features within datasets (Chen et al., 2020; Chen & He, 2021; Zbontar et al., 2021). Given that supervised learning requires substantial amounts of annotated data, and medical data collection and annotation can be prohibitively time-consuming and expensive, SSL presents an attractive alternative. Contemporary SSL methods for EEG signals typically employ pretext tasks with separate encoders to generate learning objectives (Liu et al., 2023; Guetschel et al., 2024; Weng et al., 2025). Invariance-based methods aim to optimize encoders that produce similar embeddings for multiple views of the same EEG time series (Eldele et al., 2021; Mohsenvand et al., 2020), while self-prediction methods involve masking or corrupting parts of the input data and training models through self-prediction or reconstruction (Banville et al., 2021; Cui et al., 2024). Nonetheless, these methods tend to overlook spatial relationships between EEG electrodes, and our framework is motivated to design a channel-aware pretext task based on spatial coherence predictions.

EEG oscillations within specific frequency bands often reflect similar underlying physiological processes. Although many previous models did not explicitly select frequency bands, their architectures implicitly encode preferences for certain frequency ranges (Abello et al., 2021; Molina et al., 2024). This observation motivates our use of both a band-specific approach and a band-mixture design, particularly across higher frequencies. Higher-frequency components exhibit more localized and specific connectivity patterns that quickly attenuate with distance (Łęski et al., 2013; Amo et al.,

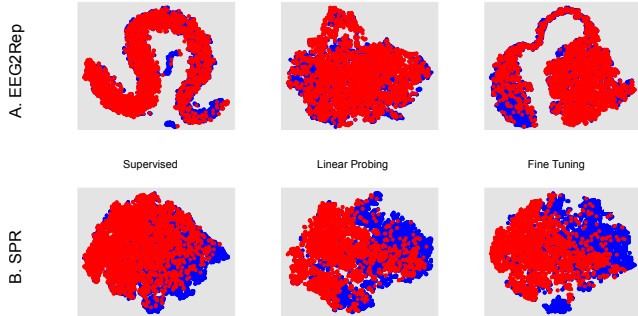

Figure 1: Comparison of t-SNE visualizations between EEG2Rep and SPR (our method) on the STEW dataset (Lim et al., 2018a). Blue and red points represent the two classes: low workload and high workload conditions, respectively. **(A)** EEG2Rep representations (top row). **(B)** SPR representations (bottom row). Our method demonstrates superior class separability with more distinct clustering patterns. Data samples were preprocessed as detailed in Appendix A.1.

2017; Fitzgerald & Watson, 2018; Arnulfo et al., 2020), whereas lower frequencies spread more broadly and are visible in large areas of the scalp. In particular, coherence decays with distance, and higher frequencies show steeper falloffs, indicating that higher-frequency activities are more local while lower-frequency activities support longer-range interactions (Jia et al., 2011; Buzsáki & Schomburg, 2015). In our pre-training, we focus on shorter-range interactions and aim to capture informative local spatial relationships by teaching the model the coherence between channel pairs in higher-frequency bands.

To support our design choices, we present t-distributed stochastic neighbor embedding (t-SNE) visualizations in Fig. 1. The visualization clearly demonstrates that our method generates more visually separable representations compared to the state-of-the-art (SOTA) EEG2Rep (Mohammadi Foumani et al., 2024) approach, indicating the creation of more meaningful and robust representations through our SSL framework. Notably, fine-tuning with our framework also exhibits superior class separability compared to supervised methods, demonstrating the enhanced performance achieved through SSL.

In summary, our contributions are as follows: **(A)** We propose the *Spatiality Preserving Representation (SPR) Learning* framework. EEG signals provide crucial input for early disease detection, and our SPR representation captures the one-to-one electrode relationship useful to functional connectivity analysis and disease localization. **(B)** Our SSL pretext task introduces a novel *band-mixture coherence approach* that captures a meaningful spatial structure of the electrode arrangements of EEG. This pretext task also does not require the keeping of separate encoders, fundamentally differing from existing invariance-based and self-prediction methods. **(C)** Despite employing a compact model architecture, our method consistently achieves *SOTA performance* through straightforward in-domain pre-training (4.7%, 9.7%, 1.6%, and 15.6% fine-tuning improvements over different datasets), demonstrating promising directions for future research.

## 2 RELATED WORK

SSL methods in EEG analysis offer substantial advantages in addressing the persistent challenges associated with acquiring high-quality labeled EEG datasets, and the field has recently witnessed significant advancement. BENDR learns representations of raw data signals through reconstruction (Kostas et al., 2021). MAEEG learns to reconstruct the masked EEG features using a transformer architecture (Chien et al., 2022). TS-TCC transforms the data into two different views and uses temporal contrasting to learn robust temporal features (Eldele et al., 2021). TF-C embeds a time-based neighborhood of an example close to its frequency-based neighborhood (Zhang et al., 2022). BIOT is a cross-data learning framework that can be used with EEG signals and SSL by tokenizing the signals into unified sentences (Yang et al., 2023). TFM-Tokenizer transforms EEG signals into a sequence of discrete, well-represented tokens in capturing critical time-frequency features from a single channel (Pradeepkumar et al., 2025). EEG2Rep learns to predict masked input in latent

representation space (Mohammadi Foumani et al., 2024). Compared to those methods, SPR does not contrast different views of the input signal or keep separate encoders, but rather adopts novel spatial coherence targets that can be precomputed efficiently.

Foundation models represent large-scale, pre-trained neural network architectures that serve as versatile backbone systems for a wide array of downstream tasks. Motivated by significant advances in natural language processing (NLP) and computer vision (CV) (Caron et al., 2021; Radford et al., 2021; Bao et al., 2022; Peng et al., 2022; Wang et al., 2023), foundation models have attracted considerable interest in recent years. Recently, LaBraM and NeuroLM have used vector-quantized encoders that establish strong conceptual and methodological connections to their CV counterparts (Jiang et al., 2024; 2025), demonstrating the transferability of foundation model concepts across different data modalities. Other notable works include EEGFormer (Chen et al., 2024), EEGM2 (Hong et al., 2025), LEAD (Wang et al., 2025), Neuro-GPT (Cui et al., 2024), EEGPT (Wang et al., 2024a), CBraMod (Wang et al., 2024b). In particular, our SPR model is not similar to those cross-domain foundation models, but rather to preserve and capture useful spatial representation through in-domain pre-training.

## 3 METHOD

Our study introduces a novel approach for generating coherence-based pseudolabels from EEG data, which involves computing band-limited magnitude-squared coherence across all possible channel pairs. This process allows for the extraction of frequency-specific connectivity patterns within predefined EEG frequency bands, which are subsequently converted into mixed probabilistic pseudolabels for downstream machine learning tasks.

### 3.1 BAND-LIMITED SPECTRAL ANALYSIS

To focus on physiologically relevant oscillatory interactions, we employ a band-limited spectral analysis rather than computing coherence across the entire frequency spectrum. This is akin to a periodogram with averaging but with an important distinction: it performs frequency-bin averaging within a specific band, unlike the full-spectrum approach. The process begins by applying the Fast Fourier Transform (FFT) to the input time series $\mathbf{x}$ to obtain its frequency-domain representation:

$$\mathbf{X}_{fft} = \text{FFT}(\mathbf{x}) \tag{1}$$

Frequency bins corresponding to the target band were selected using a frequency mask:

$$M(f) = \begin{cases} 1 & \text{if } f_{low} \leq |f| \leq f_{high} \\ 0 & \text{otherwise} \end{cases} \tag{2}$$

Here, $f_{low}$ and $f_{high}$ define the boundaries of the frequency band, and $f$ is the frequency axis, computed as $f = \frac{k \cdot f_s}{N}$ for $k = 0, 1, \ldots, N-1$, where $f_s$ is the sampling frequency and $N$ is the number of samples.

Our band-limited approach differs significantly from the traditional Welch's method for spectral density estimation, as summarized in Table 1. Welch's method enhances variance reduction by dividing the signal into overlapping segments, applying a windowing function (e.g., Hanning) to each segment, and then averaging the periodograms across these segments. In contrast, our approach processes the signal as a single time window and later uses a band-mixture approach to trade a small bias for reduced variance. More details can be found in Appendix C.

### 3.2 COHERENCE PSEUDOLABELS

Cross-power spectral density (CPSD) is a complex-valued measure that quantifies the statistical relationship between two signals or two channels of a multi-channel signal in the frequency domain. The magnitude of the CPSD at a given frequency indicates the shared power between the two signals, and a high magnitude signifies a strong correlation of power at that specific frequency. Meanwhile, the phase of the CPSD at a given frequency represents the phase difference or time lag between the two signals' components.

Table 1: Comparison of estimation methods

|  | **Traditional Welch's** | **Band-limited** |
|---|---|---|
| **Segmentation** | Overlapping time windows | Single time window |
| **Averaging** | Across time segments | Across frequency bins |
| **Resolution** | Time-frequency trade-off | Fixed by band selection |
| **Computation** | Higher (multiple segments) | Lower (single FFT) |
| **Specificity** | Full spectrum | Band-specific |

We compute the CPSD for all possible pairs of channels from the input EEG data. For any two channels, $i$ and $j$, the CPSD, $S_{ij}(f)$, is defined as:

$$S_{ij}(f) = X_i(f) \cdot X_j^*(f) \tag{3}$$

This resulted in a matrix of dimensions $\mathbf{S} \in \mathbb{C}^{C \times C \times N_{freq}}$, where $C$ is the number of channels. Each element is a complex number representing the CPSD between channel $i$ and channel $j$.

For our coherence calculation, we take the average of these complex values across the frequency bins within the specified band. Each element in the resulting matrix $\tilde{\mathbf{S}} \in \mathbb{C}^{C \times C}$ entails the average of shared power and phase difference between channel $i$ and $j$ over the entire frequency band.

The diagonal elements of $\tilde{\mathbf{S}}$, where $i = j$, represent the average power spectral density (PSD) for each channel. By extracting the diagonal from $\tilde{\mathbf{S}}$, we obtain the values corresponding to the total average power of a specific channel across the entire frequency band. The magnitude-squared coherence, $\gamma_{ij}^2$, is thus defined by:

$$\gamma_{ij}^2 = \frac{|\tilde{S}_{ij}|^2}{\tilde{S}_{ii} \cdot \tilde{S}_{jj}} \tag{4}$$

Here, the numerator corresponds to the squared magnitude of the average CPSD, and the denominator is the product of the average PSDs of channel $i$ and $j$, effectively serving as a normalization factor. This approach yields a single coherence value per channel pair for each frequency band, capturing the overall connectivity strength within that physiological frequency range, as illustrated in Fig. 2.

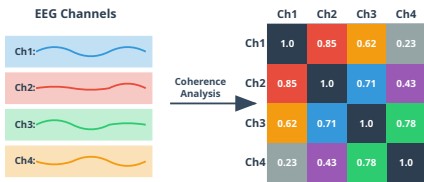

Figure 2: An example of the coherence matrix for a four-channel EEG signal. Each element represents the normalized connectivity between channel $i$ and channel $j$.

To convert the derived coherence matrices into probabilistic pseudolabels for our downstream learning task, we apply softmax normalization with a temperature parameter $\tau$ to control the sharpness of the probability distribution. The pseudolabel for each channel pair is computed as:

$$\mathbf{L}_{ij} = \frac{\exp(\gamma_{ij}^2/\tau)}{\sum_{k=1}^{C} \exp(\gamma_{ik}^2/\tau)} \tag{5}$$

The resulting pseudolabels have dimensions $C \times C$, where each element represents the normalized pseudolabel for a specific channel pair. This provides a channel-wise probability distribution over all possible connections for the frequency band, which itself can be used as a soft target.

### 3.3 BAND MIXTURE

To capture a comprehensive representation of higher-frequency connectivity, we aggregate the pseudolabels from the $\alpha$ (8-13 Hz), $\beta$ (13-30 Hz) and $\gamma$ (30-100 Hz) bands, excluding lower-frequency $\delta$ (0.5-4 Hz) and $\theta$ (4-8 Hz) bands. The $\alpha$, $\beta$ and $\gamma$ oscillations are robustly associated with active cognitive processes, conscious perception, and local or inter-regional communication during wakeful states (Başar et al., 2001; Newson & Thiagarajan, 2019). In particular, $\alpha$ rhythms are linked to attentional states and sensorimotor integration, $\beta$ to motor control and higher-level cognitive processing, and $\gamma$ to feature binding and conscious awareness.

At the center of our interest are spatial relationships. Higher frequency oscillations typically exhibit more spatially localized patterns compared to the widespread, diffuse nature of $\delta$ and $\theta$ rhythms (Yang et al., 2020). This property makes $\alpha$, $\beta$ and $\gamma$ bands more suitable for distinguishing fine-grained connectivity patterns between specific channel pairs. The chosen higher-frequency oscillations are also less affected by artifacts, whereas lower frequency bands are highly susceptible to physiological artifacts, such as eye movements and muscle activity, which can confound connectivity analysis.

For our method, the final coherence pseudolabel $\mathbf{L}_{ij}$ is computed as the average of the pseudolabels from the $\alpha$, $\beta$, and $\gamma$ bands:

$$\mathbf{L}_{ij} = \frac{1}{N_b} \sum_{k \in \mathcal{B}} \mathbf{L}_{ij,k} \tag{6}$$

Here, $\mathcal{B} = \{\alpha, \beta, \gamma\}$, $N_b = 3$, and $\mathbf{L}_{ij,k}$ is the coherence-based pseudolabel for the connection between channel $i$ and $j$ in the frequency band $k$. The resulting $\mathbf{L}_{ij}$ represents the integrated coherence pseudolabel for the three selected physiological bands. Each band is weighted equally to ensure a balanced contribution to the final representation.

This band-mixture coherence approach provides a stable and interpretable measure of connectivity across functionally relevant frequency ranges, which is well-suited for providing pseudolabels for the SSL pretext task. It forces the model to learn the underlying inter-channel relationships and spatial-temporal patterns inherent to EEG data, a task more aligned with downstream cognitive analysis than low-level signal reconstruction.

## 4 MODEL ARCHITECTURE

Our proposed model architecture is a two-stage transformer-based framework designed for EEG signal analysis, comprising both an SSL pre-training stage and a supervised fine-tuning stage. An illustration of the model architecture is shown in Fig. 3. It maintains a strict one-to-one relationship between input channels and the channels of learned representations, which is crucial for interpretable EEG analysis.

### 4.1 SELF-SUPERVISED PRE-TRAINING

The pre-training stage takes input a multi-channel EEG sequence $\mathbf{X} \in \mathbb{R}^{C \times T}$, where $C$ is the number of channels and $T$ is the number of time points, the signal is first divided into non-overlapping temporal patches. Each patch of length $P$ is treated as a token. For an input of $T = 2000$ and $P = 125$, this yields 16 tokens per channel, resulting in an initial sequence of $16 \times C$ tokens. The sequence is passed through a linear embedding layer that projects each patch into an embedding space.

Since we are interested in the spatial coherence of whole sequences instead of time segments, a continuous masking strategy is applied, where a subset of approximately 30% of patches is masked, followed by the addition of positional encodings to preserve the temporal order. These embeddings are then processed by a lightweight context encoder, which consists of four transformer blocks, to generate contextualized representations of the visible signal segments. A cross-attention predictor, composed of two transformer blocks, is then used to predict the masked representations. Here, learnable mask tokens act as queries, while the output features from the context encoder serve as

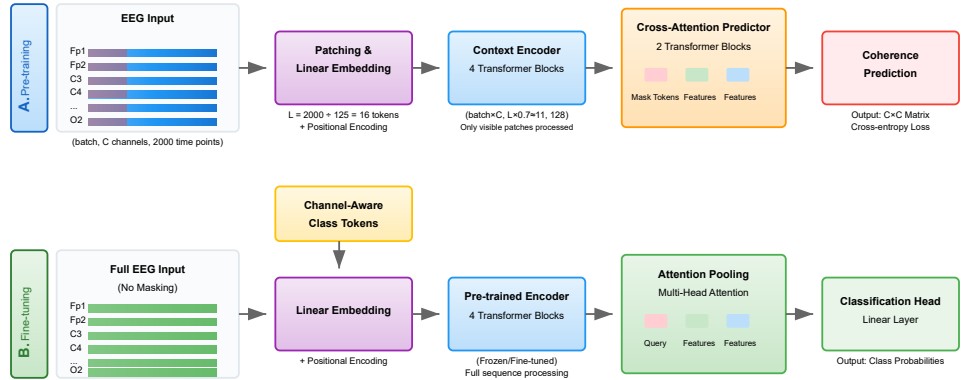

Figure 3: Two-stage SPR architecture with masked pre-training and supervised fine-tuning. **(A)** pre-training stage: A raw $C$-channel EEG signal (e.g., with channel Fp1, Fp2, ..., O2, and 2000 time points) is temporally patched and continuously masked. The starting indices of masks are the same cross channels but randomly drawn for different batches. Visible patches are processed by a four-block context encoder. A cross-attention predictor then uses mask tokens as queries to predict the masked features, leading to final coherence patterns. **(B)** Fine-tuning stage: Complete EEG signals are processed alongside $C$ channel-aware class tokens. The outputs from the pre-trained encoder are aggregated via attention pooling. Finally, a linear head outputs task-specific predictions.

both keys and values. This design allows the model to predict the content of the masked segments based on the surrounding temporal context.

Crucially, the pre-training objective is not signal reconstruction but rather *coherence prediction*. The model input is partially masked, but targets are computed from the unmasked signal. The model predicts a $C \times C$ coherence matrix that is optimized with a cross-entropy loss.

### 4.2 SUPERVISED FINE-TUNING

In the fine-tuning stage, the masking strategy is removed, allowing the model to process complete, unmasked EEG sequences. The input undergoes the same patching and linear embedding process as in pre-training, ensuring consistency with the learned representations. The pre-trained context encoder can either be frozen (linear probing) or fine-tuned.

A key feature in this stage is the introduction of *channel-aware class tokens*. The encoder processes the complete sequence of patch tokens and channel-aware class tokens, where the latter are $C$ distinct learnable tokens, each corresponding to a specific EEG channel. An attention-based pooling mechanism is then applied, where the queries are learned and the encoder's output features serve as keys and values. This attention-based pooling allows the model to selectively aggregate information into channel-weighted representations. Finally, a linear layer maps the channel-weighted representations to the class probability output.

## 5 DATASETS

We evaluate our approach on four publicly available EEG datasets spanning different applications and acquisition settings. The **Simultaneous Task EEG Workload (STEW)** dataset contains 14-channel recordings from 48 participants performing multitasking workload experiments, with binary classification labels for high/low mental workload (Lim et al., 2018b). From the Temple University Hospital (TUH) EEG Corpus, we utilize two subsets: **TUH Abnormal EEG Corpus (TUAB)** for binary normal/abnormal EEG classification, and **TUH EEG Events (TUEV)** for event detection across six categories (Obeid & Picone, 2016). To compare the performances with foundation models, we also include experimental results for seizure detection on the **Children's Hospital Boston-MIT (CHB-MIT)** database (Shoeb, 2009; Goldberger et al., 2000).

These datasets exhibit diverse characteristics in terms of channel configuration (14-16 channels), sampling rates (128-256 Hz), and temporal segments (2-10 seconds), providing a comprehensive evaluation framework. Dataset statistics are summarized in Table 2, with detailed preprocessing and acquisition protocols described in Appendix A.

Table 2: Dataset overview

| Dataset | Rate | #Channels | Length | #Samples | Task |
|---------|------|-----------|--------|----------|------|
| STEW | 128Hz | 14 | 2s | 28,512 | Mental workload |
| TUAB | 200Hz | 16 | 10s | 409,455 | Abnormal classification |
| TUEV | 250Hz | 16 | 5s | 113,353 | Event detection |
| CHB-MIT | 256Hz | 16 | 10s | 326,991 | Seizure detection |

## 6 RESULTS

We present our main results in Table 3 and 4, which demonstrate the performance of our proposed SPR method under linear probing and fine-tuning evaluation protocols, respectively. The reported baseline metrics are consistent with those in EEG2Rep (Mohammadi Foumani et al., 2024). Each entry represents the mean value and standard deviation computed over five independent random seeds to ensure statistical reliability.

Our SPR method consistently achieves SOTA performance across all datasets and evaluation metrics and demonstrates substantial improvements compared to the second-best method under the linear probing protocol. We observe gains of 4.9% in accuracy and 12.7% in AUROC for STEW. There are improvements of 11.4% in balanced accuracy and 1.7% in weighted F1 score on TUEV, and improvements of 2.5% in accuracy and 4.7% in AUROC on TUAB.

The fine-tuning results further validate the effectiveness of our approach, with notable improvements compared to the second-best method across all datasets. We observe gains of 4.7% in accuracy and 11.9% in AUROC on STEW for our pre-trained model. There are also improvements of 9.7% in balanced accuracy and 3% in weighted F1 score on TUEV, and improvements of 1.6% in accuracy and 1% in AUROC on TUAB.

The consistent performance gains across both evaluation protocols indicate that our learned representations are both semantically meaningful and amenable to task-specific adaptation. The fact that our randomly initialized model serves as a strong baseline indicates that our contributions extend beyond simply learning better representations to encompassing a more effective overall approach to EEG analysis. Meanwhile, our pre-trained SPR model establishes new SOTA results by substantial margins. The consistent performance gains across diverse datasets with different characteristics further underscore the generalizability and robustness of our method.

Table 3: Linear Probing Performance (%) of SPR in comparison to other methods

| Method | STEW | | TUEV | | TUAB | |
|--------|------|------|------|------|------|------|
| | Accuracy | AUROC | B-Accuracy | W-F1 | Accuracy | AUROC |
| BIOT | 67.5(2.1) | 67.7(3.6) | 40(1.9) | 66(2) | 75.1(2.8) | 82.9(2) |
| BENDR | 63(1.1) | 63(1.1) | 37.4(3.1) | 61.3(3.2) | 72.8(4.2) | 79.9(4.1) |
| MAEEG | 68(1.9) | 68.6(1.9) | 37.2(3) | 61.4(3.1) | 72.6(4) | 79.8(4.1) |
| TS-TCC | 64.5(1.6) | 64.6(1.7) | 36(2.9) | 60.7(3) | 74.4(3.1) | 81(3) |
| TF-C | 58.8(2.4) | 58.7(2.4) | 30.1(4.1) | 56.2(4.1) | 69.3(5.8) | 75.8(3.8) |
| EEG2Rep | 69(1) | 69.1(1.2) | 43.3(3.1) | 70(3.2) | 76.6(3.3) | 83.2(3.3) |
| **SPR** | **73.9(1.5)** | **81.8(2.1)** | **54.7(4.3)** | **71.7(3.5)** | **79.1(0.5)** | **87.9(0.5)** |

### 6.1 ABLATIONS

An important observation from our experiments is that loss function selection significantly impacts performance on highly imbalanced datasets such as TUEV. Given this class imbalance, we employed

Table 4: Fine-tuning performance (%) of SPR in comparison to other methods

| Method | STEW | | TUEV | | TUAB | |
|---|---|---|---|---|---|---|
| | Accuracy | AUROC | B-Accuracy | W-F1 | Accuracy | AUROC |
| BIOT | 69.9(2.2) | 70.1(2.6) | 46(1.7) | 70(2) | 79.2(2.2) | 87.4(2) |
| BENDR | 69.7(2.1) | 69.8(2) | 41.2(2.9) | 67.3(3) | 77(4) | 84(3.4) |
| MAEEG | 72.5(3.7) | 72.5(3.2) | 41.2(3.7) | 67.4(3.7) | 77.6(3.6) | 86.6(3.3) |
| TS-TCC | 71(3) | 71(3) | 41(2.6) | 68.7(2.9) | 79.7(3) | 87(2.7) |
| TF-C | 68.7(1.1) | 68.7(1.8) | 40.1(3.7) | 66.2(3.9) | 72.3(5.6) | 78.5(3.9) |
| EEG2Rep | 73.6(1.5) | 74.4(1.5) | 53(1.6) | 75.1(1.2) | 80.5(2.2) | 88.4(3.1) |
| SPR(Random) | 75.3(0.9) | 83.0(0.8) | 58.1(2.1) | 75.5(2.7) | 79.2(0.3) | 86.9(0.3) |
| **SPR (Pre-train)** | **78.3(1.9)** | **86.3(2.5)** | **62.7(1.7)** | **78.1(2.4)** | **82.1(0.2)** | **89.4(0.3)** |

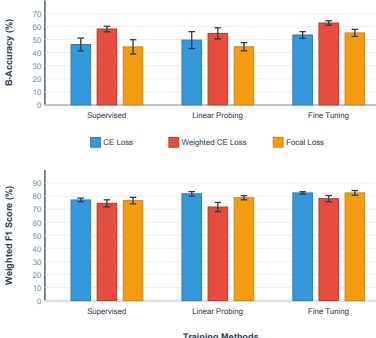

Figure 4: Evaluation of different loss functions on the TUEV dataset. The top subplot shows balanced accuracy performance, and the bottom subplot shows weighted F1 score performance.

weighted cross-entropy loss for supervised learning on TUEV, but it is crucial to examine how different loss function choices affect our model's performance across various metrics. Fig. 4 reveals distinct patterns in loss function effectiveness. Weighted cross-entropy loss achieves the highest balanced accuracy while maintaining competitive performance on weighted F1 score, though the improvement in F1 is modest. Interestingly, when focusing specifically on fine-tuning results, focal loss demonstrates an optimal balance between balanced accuracy and weighted F1 score, suggesting its effectiveness for task-specific adaptation on imbalanced data.

These findings align with our main results and highlight the trade-off in loss function selection. While weighted cross-entropy loss yields significantly higher balanced accuracy, alternative loss functions can achieve substantial improvements in weighted F1 score, albeit with some compromise in balanced accuracy. Importantly, our SPR method demonstrates consistent improvements across all loss function choices, indicating the robustness of our approach.

To provide a comprehensive assessment of each key component's contribution, we present the averaged accuracy across linear probing and fine-tuning protocols on the STEW dataset in Table 5, which is obtained over five random seeds. In particular, the most substantial performance degradation (4.2% accuracy drop) occurs when positional encoding is removed, and the removal of channel-aware class tokens also results in a 1.7% accuracy reduction. These significant declines underscore the importance of maintaining spatial and temporal relationships within the learned representations, as positional encoding and channel-aware class tokens enable the model to distinguish between identical signal patterns occurring at different time points and electrode locations, providing strong empirical support for our hypothesis that the framework effectively captures spatial-temporal dynamics in EEG signals.

We observe that excessively large $\tau$ values significantly degrade performance. Performance deterioration with overly high masking ratios indicates that excessive signal occlusion prevents the model from extracting meaningful coherence features. This suggests an optimal balance where sufficient signal information remains visible to infer spatial-temporal relationships while still providing

Table 5: Key component analysis on average performance (%)

| Component | Choice | Accuracy (**Base: 76.1**) | Change |
|---|---|---|---|
| Temperature ($\tau$) | 0.8 | 75.5 | ↓0.6 |
| | 1.0 | 75.7 | ↓0.4 |
| | 1.3 | 75.2 | ↓0.9 |
| | 1.5 | 74.0 | ↓2.1 |
| Masking ratio | 0.1 | 75.6 | ↓0.5 |
| | 0.2 | 75.0 | ↓1.1 |
| | 0.5 | 74.8 | ↓1.3 |
| | 0.7 | 73.9 | ↓2.2 |
| Pseudolabel | Phase synchronization | 75.4 | ↓0.7 |
| | Temporal correlation | 75.9 | ↓0.2 |
| | Welch's method | 74.9 | ↓1.2 |
| | No band limit (All Hz) | 75.8 | ↓0.3 |
| | Single band (8–100 Hz) | 74.8 | ↓1.3 |
| | Remove diagonal | 74.5 | ↓1.6 |
| Model component | No attention pool | 75.5 | ↓0.6 |
| | No channel-aware class token | 74.4 | ↓1.7 |
| | No layer normalization | 75.9 | ↓0.2 |
| | No positional encoding | 71.9 | ↓4.2 |
| | No mask token | 75.6 | ↓0.5 |
| | Random masking | 74.8 | ↓1.3 |

adequate reconstruction challenge for effective learning. Meanwhile, our spatial coherence computation demonstrates superior performance even when diagonal elements are retained. As detailed in Appendix B, the model successfully learns spatial relationships despite strong diagonal coherence values.

## 6.2 COMPARISON TO FOUNDATION MODELS

We provide an additional comparison in terms of fine-tuning performance to recently developed models in Table 6, in which we compare our SPR model with both non-foundational models including EEGNet (Lawhern et al., 2018), EEGConformer (Song et al., 2022), SPaRCNet (Jing et al., 2023), CNN-Transformer (Peh et al., 2022), ST-Transformer (Song et al., 2021), and foundation models including BIOT (Yang et al., 2023), CBraMod (Wang et al., 2024b), LaBraM(Jiang et al., 2024). We can observe that the balanced accuracy on TUAB of our method is significantly higher than all other non-foundational models, and is comparable to LaBraM, a much larger foundation model. For the seizure detection task on CHB-MIT, we achieve the highest balanced accuracy and AUROC by a large margin (15.6% and 5.2% higher than the second best model CBraMod), possibly indicating a more useful technique to pre-train the model for the seizure detection task based on local-connectivity patterns.

To explain the CHB-MIT results, we examine the neuroanatomy more closely via visualizations (Fig. 7D and 8 in Appendix B) and observe structured activation patterns that highlight functionally connected brain regions that can help identify focal epilepsy (Watanabe, 1989; Plummer et al., 2008; Foldvary et al., 2001). In Fig. 8, there is more pronounced local activation over **frontal** and **prefrontal** areas, as well as the **temporal** lobe in the $\beta$ and $\gamma$ bands, and these regions exhibit correspondingly elevated local connectivity in Fig. 7D. In contrast, central and parietal regions show relatively weak activation and limited local connectivity in the learned heat map. The learned heat map also reveals dominant local connectivity rather than long-range interactions, consistent with our design choice to emphasize local connectivity through higher-frequency coherence.

In Table 7, we further demonstrate the results evaluated on TUAB with cross-domain pre-training. With adding less than 15% pre-train samples, the standard deviation decreases considerably, indicating better generalization capability, and the balanced accuracy increases by 0.4%. This improved performance indicates the robustness and generalization capability of our model across datasets.

Table 6: Fine-tuning performance (%) comparisons with foundation models

| Method | Params | TUAB | | CHB-MIT | |
|---|---|---|---|---|---|
| | | B-Accuracy | AUROC | B-Accuracy | AUROC |
| EEGNet | 0.003M | 76.4(0.4) | 84.1(0.3) | 56.6(1.1) | 80.5(1.4) |
| EEGConformer | 0.55M | 77.6(0.5) | 84.5(0.4) | 59.8(1.4) | 82.3(1.7) |
| SPaRCNet | 0.79M | 79(0.2) | 86.8(0.1) | 58.8(1.9) | 81.4(1.5) |
| CNN-Transformer | 3.2M | 77.8(0.2) | 84.6(0.1) | 63.9(0.7) | 86.6(0.8) |
| ST-Transformer | 3.5M | 79.7(0.2) | 87.1(0.2) | 59.2(2) | 82.4(4.9) |
| BIOT | 3.2M | 79.6(0.6) | 88.2(0.4) | 70.7(4.6) | 87.6(2.8) |
| CBraMod | 4M | **82.9(0.2)** | **92.3(0.1)** | 74(2.8) | 88.9(1.5) |
| LaBraM | 5.8M | 81.4(0.2) | 90.2(0.1) | 70.8(3.6) | 86.8(2) |
| SPR (Random) | 0.8M | 78.7(0.3) | 86.9(0.3) | 54.7(1.2) | 76.3(4.1) |
| SPR (In-domain) | 0.8M | 81.3(0.6) | 89.4(0.3) | **89.6(0.8)** | **94.1(1.4)** |

Table 7: Fine-tuning performance (%) of SPR on TUAB with cross-domain pre-training

| Dataset | #Samples | B-Accuracy | AUROC |
|---|---|---|---|
| TUAB (Pre-train) | 372,510 | 81.3(0.6) | 89.4(0.3) |
| + TUEV | 425,873 | 81.7(0.2) | 89.4(0.3) |

## 7 DISCUSSION

The results presented in Table 3 and 4 provide evidence for the effectiveness of our SPR framework, demonstrating that our approach not only successfully addresses fundamental challenges in EEG representation learning, but also establishes new performance benchmarks across multiple evaluation protocols and datasets. Our approach introduces a novel time-frequency self-prediction paradigm that combines spatial coherence prediction with implicit reconstruction through masking. The ablation studies further validate our architectural choices and substantiate our hypothesis that effective EEG analysis requires explicit modeling of channel-specific characteristics, as different electrode locations capture distinct neural activities that must be processed with spatial awareness.

The computational complexity of coherence calculation is overall $\mathcal{O}(C^2 N_{freq})$ as the FFT computation $\mathcal{O}(CT \log T)$ is typically smaller for segmented EEG signals ($T \log T \ll C N_{freq}$), where $N_{freq}$ is the number of frequency bins. While the quadratic scaling with the number of channels might seem concerning for high-density EEG systems, the computational burden remains manageable since coherence pseudolabels can be precomputed only once per sample. For extremely high-density channel systems, channel subgrouping strategies can be attractive.

Our band-mixture coherence approach provides a single connectivity value per channel pair within specified physiological frequency bands, effectively summarizing spectral relationships into lower-dimensional representations suitable for deep learning applications, and reducing variance while capturing essential connectivity patterns. However, we acknowledge that averaging information across frequency ranges may occasionally obscure fine-grained frequency-specific relationships. The trade-off between frequency resolution and representation dimensionality represents a conscious design decision that prioritizes robust learning over spectral detail.

Although our primary motivation is to design a spatiality preserving EEG representation learning framework rather than developing a general foundation model, comprehensive comparisons with existing foundation models demonstrate the competitive advantages of our SPR framework. Future research directions include exploring cross-domain pre-training capabilities of the framework, investigating adaptive frequency band selection mechanisms, and extending the approach to other neurophysiological signal modalities. The demonstrated effectiveness of our spatiality preserving paradigm also opens new avenues for developing more sophisticated neural signal representation learning methods that explicitly model the complex spatial-temporal dynamics inherent in brain activities.

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

# A  DETAILED DATASET DESCRIPTION AND PREPROCESSING

## A.1  SIMULTANEOUS TASK EEG WORKLOAD (STEW)

The STEW dataset comprises raw EEG recordings from 48 participants performing multitasking workload experiments using the SIMKAP multitasking test. Each experimental session began with baseline resting-state recordings to establish individual neural activity baselines prior to task engagement. EEG signals were acquired using a 14-channel Emotiv EPOC headset at 128 Hz sampling frequency, providing 2.5 minutes of continuous recording per participant. Following each experimental stage, participants provided subjective mental workload assessments using a 9-point scale (1 = very low workload, 9 = very high workload).

**Preprocessing:** EEG signals were segmented into 2-second non-overlapping epochs, yielding a total of 28,512 samples. The dataset followed a standardized train-validation-test partition: 17,820 training samples, 3,564 validation samples, and 7,128 test samples. For self-supervised pre-training, we combined the training and validation sets (21,384 samples in total) to maximize the effectiveness of representation learning. Binary classification labels were derived by thresholding subjective ratings, with scores $> 4$ categorized as "high workload" and scores $\leq 4$ as "low workload". We followed the preprocessing method in EEG2Rep (Mohammadi Foumani et al., 2024) and applied an additional 14 bipolar montage to the dataset: "AF3-F7", "F7-T7", "T7-P7", "P7-O1", "AF4-F8", "F8-T8", "T8-P8", "P8-O2", "AF3-F3", "F3-FC5", "AF4-F4", "F4-FC6", "F3-F4", and "FC5-FC6".

**Access:** STEW is publicly accessible through IEEE DataPort at `https://dx.doi.org/10.21227/44r8-ya50` (Lim et al., 2018a).

## A.2  TEMPLE UNIVERSITY HOSPITAL EEG CORPUS

### A.2.1  TUH ABNORMAL EEG CORPUS (TUAB)

The Temple University Abnormal EEG Corpus (TUAB) represents a curated subset of the larger Temple University Hospital EEG dataset, specifically comprising clinical recordings with abnormal neurological findings. This clinical dataset contains 2339 recordings and has a slightly unbalanced class distribution (42% abnormal, 58% normal) (de Diego, 2017). TUAB serves as a critical benchmark for evaluating automated neurological disorder detection systems.

**Preprocessing:** Following the preprocessing protocol established by BIOT (Yang et al., 2023), we employed a standardized 16-channel bipolar montage configuration adhering to the international 10-20 electrode placement system. The bipolar electrode pairs were organized into four chains: "FP1-F7", "F7-T3", "T3-T5", "T5-O1", "FP2-F8", "F8-T4", "T4-T6", "T6-O2", "FP1-F3", "F3-C3", "C3-P3", "P3-O1", "FP2-F4", "F4-C4", "C4-P4", and "P4-O2". Each channel underwent amplitude normalization using the 95th percentile of absolute amplitude values as the scaling factor, providing robust normalization that minimizes the influence of extreme artifacts while preserving physiologically relevant signal dynamics. EEG signals were subsequently segmented into 10-second non-overlapping epochs, yielding a total of 409,455 samples for binary normal/abnormal classification tasks. Those samples were partitioned to 299,656 training samples (73.2%), 72,854 validation samples (17.8%), and 36,945 test samples (9.0%). For self-supervised pre-training, we combined the training and validation sets to create an expanded pre-training corpus of 372,510 samples.

### A.2.2  TUH EEG EVENTS (TUEV)

The Temple University EEG Events (TUEV) corpus comprises clinically annotated EEG segments with expert-labeled neurological events across six distinct categories: (1) spike and sharp wave, (2) generalized periodic epileptiform discharges, (3) periodic lateralized epileptiform discharges, (4) eye movement, (5) artifacts, and (6) background activity. This comprehensive annotation scheme encompasses both pathological epileptiform patterns and common non-pathological events, making TUEV particularly valuable for developing robust automated EEG event detection systems in clinical settings.

**Preprocessing:** TUEV follows an identical preprocessing pipeline to TUAB, employing the same standardized 16-channel bipolar montage configuration based on the international 10-20 system. Each channel underwent robust amplitude normalization using the 95th percentile of absolute am-

plitude values. Given the shorter duration requirements for event detection tasks, EEG signals were segmented into 5-second non-overlapping epochs, producing a total of 113,353 labeled samples for multi-class event classification. The dataset exhibits inherent class imbalance typical of clinical EEG event detection scenarios, where pathological events are naturally less frequent than background activity. The dataset was partitioned into 75,859 training samples (66.9%), 8,073 validation samples (7.1%), and 29,421 test samples (26.0%). For self-supervised pre-training, we combined training and validation sets to create an expanded corpus of 83,932 samples.

**Access:** Both TUAB and TUEV datasets are accessible through the Temple University EEG Resources repository upon request at `https://isip.piconepress.com/projects/nedc/html/tuh_eeg/index.shtml`.

### A.3 THE CHILDREN'S HOSPITAL BOSTON-MIT (CHB-MIT)

The Children's Hospital Boston-MIT (CHB-MIT) database (Shoeb, 2009; Goldberger et al., 2000) represents a publicly available dataset comprising long-term EEG recordings from 24 pediatric patients (ages 3-22 years) with medically intractable epilepsy. The recordings were acquired during presurgical evaluation periods following controlled withdrawal of anti-seizure medications, enabling comprehensive characterization of spontaneous seizure patterns and assessment of surgical candidacy. This clinical protocol provides authentic seizure events in their natural temporal context, making CHB-MIT a good standard for evaluating automated seizure detection algorithms. EEG signals were recorded using scalp electrodes positioned according to the international 10-20 system, with data acquisition at 256 Hz sampling frequency to capture high-frequency seizure components. The dataset encompasses multi-day continuous recordings, providing both ictal (seizure) and extensive interictal (non-seizure) periods. Each recording session contains expert neurologist annotations marking precise seizure onset and offset times, enabling binary classification into seizure and non-seizure states with clinical-grade ground truth labels.

**Preprocessing:** Following the preprocessing methodology established by BIOT, we employed a standardized 16-channel bipolar montage configuration to ensure consistency with established benchmarks. The selected electrode pairs form four chains: "FP1-F7", "F7-T7", "T7-P7", "P7-O1", "FP2-F8", "F8-T8", "T8-P8", "P8-O2", "FP1-F3", "F3-C3", "C3-P3", "P3-O1", "FP2-F4", "F4-C4", "C4-P4", "P4-O2". Each channel underwent robust amplitude normalization using the 95th percentile of absolute amplitude values to maintain signal integrity while reducing inter-subject variability. The continuous recordings were segmented into 10-second non-overlapping epochs, yielding 326,991 labeled samples suitable for seizure detection tasks. The dataset was divided into 286,964 training samples (87.8%), 23,065 validation samples (7.1%), and 16,962 test samples (5.2%). For self-supervised pre-training, we combined training and validation sets to create an expanded corpus of 310,029 samples.

**Access:** The CHB-MIT dataset is accessible through PhysioNet at `https://doi.org/10.13026/C2K01R`.

### A.4 DATA PARTITIONING

For the STEW dataset, we implemented a subject-wise train/validation/test split to evaluate the model's cross-subject generalization capabilities and account for inter-subject variability. For TUAB and TUEV, we adhered to the inherent training/test splits provided with the datasets, which are consistent with established benchmarking protocols. Within the designated pre-training sets, we created subject-wise validation splits. For TUAB, the pre-training set was split into 80% training and 20% validation subsets. For TUEV, we divided pre-training sets into 90% training and 10% validation subsets. The CHB-MIT dataset was partitioned by subject to evaluate performance on unseen individuals. Subjects 1-20 were designated for the training set, subjects 21-22 for the validation set, and subjects 23-24 for the test set.

## B    VISUALIZATION

The visualizations on TUAB and CHB-MIT are shown in Fig. 5, 6 and 7, 8, respectively. The channels matrices contain the axis ordered by both the bipolar chains and brain regions. The * sign signals the same channel name but followed by a different referencing path.

A key observation from the coherence matrices is that despite the dominance of diagonal elements across all target labels, our model successfully learns to focus on meaningful off-diagonal spatial relationships rather than being misled by these trivially high self-coherence values. This suggests that our model effectively captures the underlying neural dynamics while mitigating potential artifacts from self-coherence terms.

The model predictions exhibit clear chain-wise and region-wise activation patterns in both Fig. 5D and 7D, providing evidence that our framework effectively captures genuine spatial relationships between electrode locations. These structured activation patterns indicate that the model learns to identify functionally connected brain regions rather than simply memorizing statistical regularities in the pre-training dataset.

When examining predictions across different clinical conditions, we observe distinct spatial relationship patterns. Normal and non-seizure subjects exhibit sparser spatial connectivity predictions, while abnormal and seizure subjects demonstrate visually denser interconnection patterns. This class-dependent variation in learned spatial representations creates discriminative features that naturally separate different underlying pathological states, thereby enhancing the utility of learned representations for downstream tasks.

Meanwhile, both ground truth labels and model predictions consistently demonstrate the well-established adjacency effect, where spatially neighboring electrodes exhibit high coherence regardless of their anatomical brain region assignments. For instance, electrode pairs T5-P3 and P4-T6 show strong coherence in the normal subject of Fig. 5A, and Fp2-F4 in the non-seizure state of Fig. 7C as well, reflecting the physical proximity of these recording sites. This neurophysiologically plausible pattern provides strong validation for using coherence as a prediction target, as it confirms that our model learns spatial relationships that align with known principles of neural signal propagation. However, we need to point out that this adjacency effect is susceptible to volume conduction effects, which can create spurious correlations between channels due to signal propagation through conductive head tissues (van den Broek et al., 1998).

The topographic maps presented in Fig. 6 and 8 reveal a clear frequency-dependent pattern in spatial specificity, providing strong empirical support for our frequency band selection strategy. Lower frequency bands ($\delta$ and $\theta$) exhibit diffuse, spatially non-specific activation patterns with limited discriminative power between electrode locations. In contrast, higher frequency bands ($\alpha$, $\beta$, and $\gamma$) demonstrate increasingly focal and spatially differentiated patterns. In particular, the $\gamma$ band here exhibits the most pronounced spatial specificity and demonstrates visually distinct topographic patterns among normal versus abnormal subjects, as well as seizure versus non-seizure states.

The enhanced spatial discriminability in the $\alpha$, $\beta$, and $\gamma$ ranges suggests that these frequencies carry more informative spatial-temporal signatures for representation learning. By prioritizing these frequency ranges, our model can take advantage of their inherent spatial specificity to learn more meaningful and diagnostically relevant representations.

## C    MATHEMATICAL FRAMEWORK

For discrete-time signals $x[n]$ of length $N$, the Discrete Fourier Transform is computed as:

$$X[k] = \sum_{n=0}^{N-1} x[n]e^{-j2\pi kn/N}, \quad k = 0, 1, \ldots, N-1 \tag{7}$$

The frequency bins correspond to $f[k] = \frac{k \cdot f_s}{N}$, where $f_s$ is the sampling frequency.

Given a frequency band defined by $[f_{low}, f_{high}]$, we define the band mask as:

$$M[k] = \begin{cases} 1 & \text{if } f_{low} \leq |f[k]| \leq f_{high} \\ 0 & \text{otherwise} \end{cases} \tag{8}$$

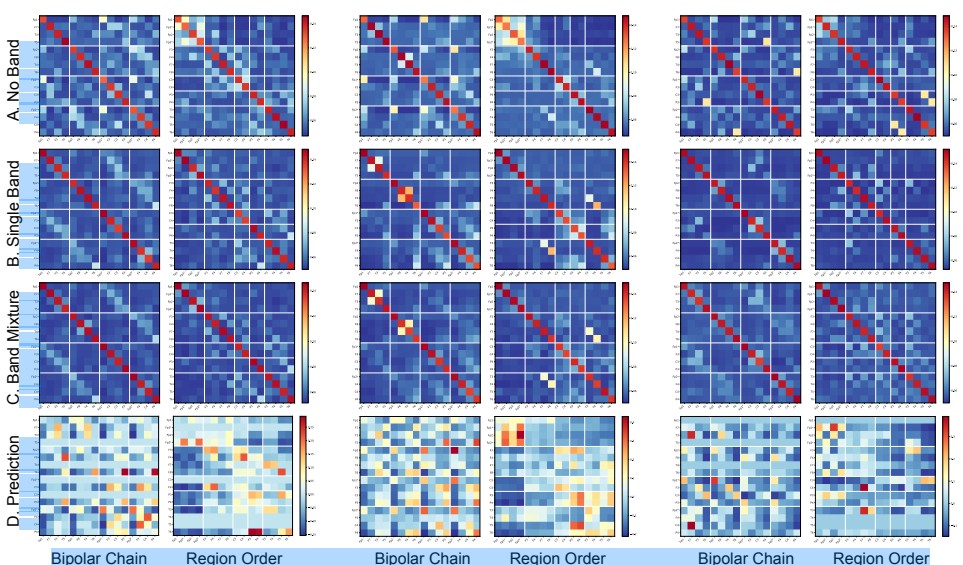

Figure 5: Channel Matrices for three different samples (abnormal, abnormal, normal) from TUAB. **(A)** Coherence labels without band-limits. **(B)** Coherence labels with single band (8–100 Hz). **(C)** Coherence labels with band mixture (8–13, 13–30, 30–100 Hz). **(D)** Coherence labels predicted by the model.

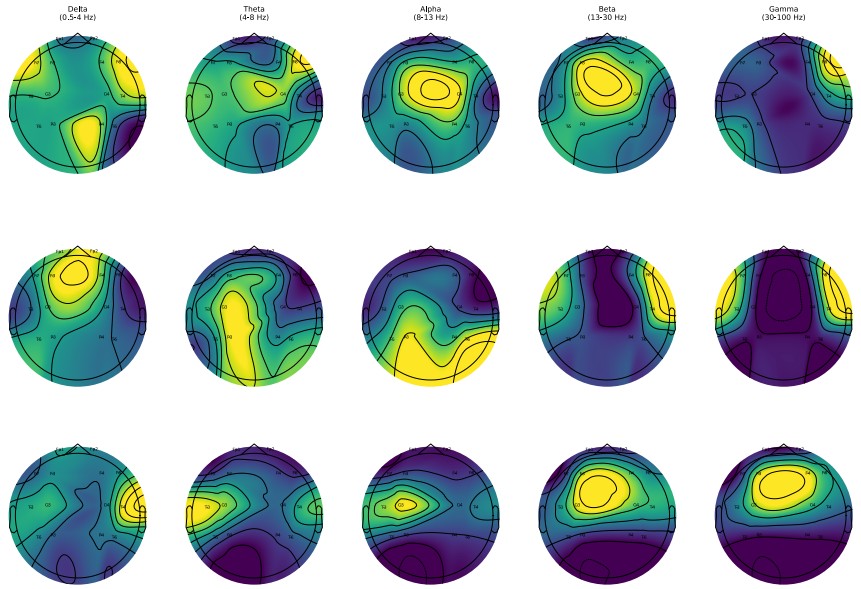

Figure 6: The topographic maps in power (dB) for three different samples (abnormal, abnormal, normal) from TUAB. The values are averaged for channels at the same position. Each sample corresponds to the sample in Fig. 5.

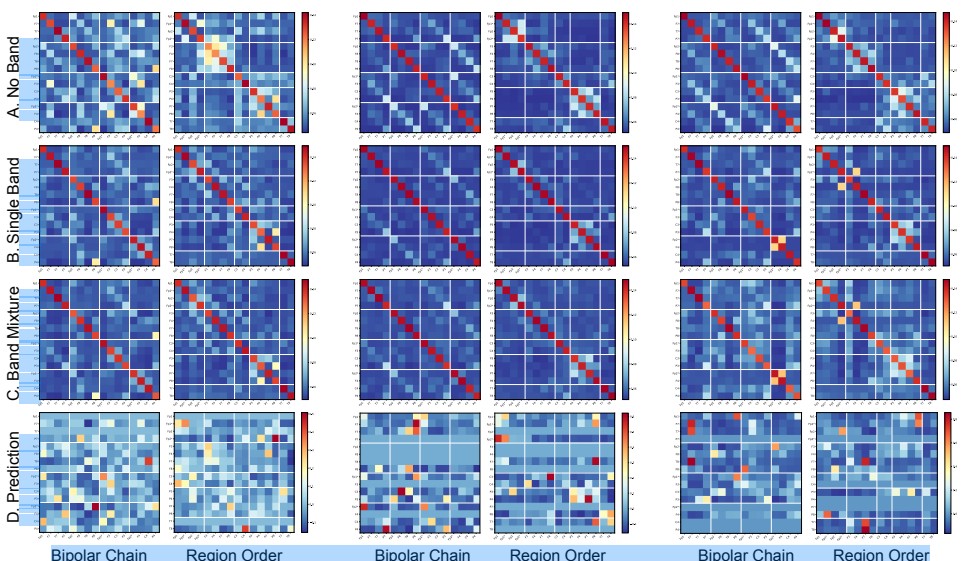

Figure 7: Channel Matrices for three different samples (seizure, seizure, non-seizure) from CHB-MIT. **(A)** Coherence labels without band-limits. **(B)** Coherence labels with single band (8–100 Hz). **(C)** Coherence labels with band mixture (8–13, 13–30, 30–100 Hz). **(D)** Coherence labels predicted by the model.

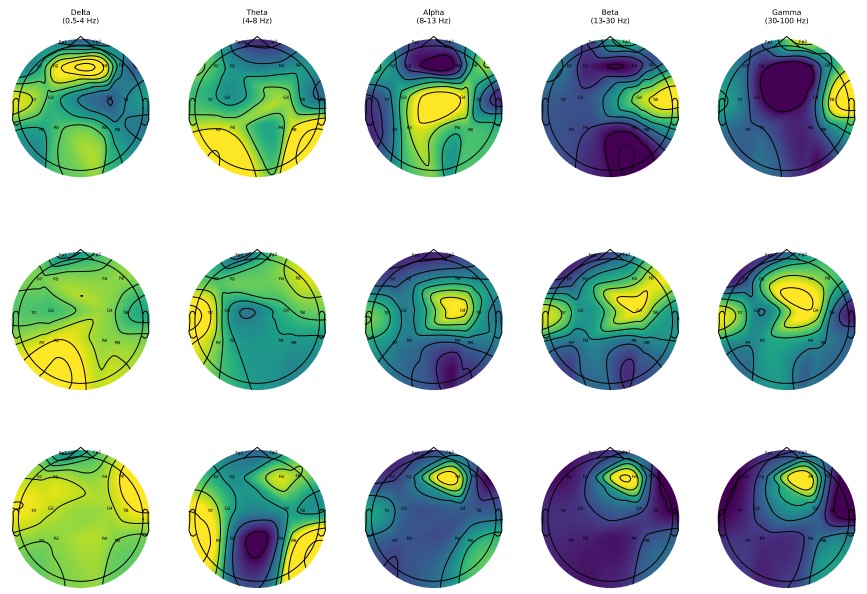

Figure 8: The topographic maps in power (dB) for three different samples (seizure, seizure, non-seizure) from CHB-MIT. The values are averaged for channels at the same position. Each sample corresponds to the sample in Fig. 7.

The band-limited cross-spectral density between channel $i$ and $j$ is:

$$\tilde{S}_{ij} = \frac{1}{N_{freq}} \sum_{k:M[k]=1} X_i[k] \cdot X_j^*[k] \tag{9}$$

Here, $N_{freq}$ is the number of frequency bins in the band.

The coherence function is formally defined as the ratio of the CPSDs to the product of the individual PSDs:

$$\gamma_{xy}^2(f) = \frac{|S_{xy}(f)|^2}{S_{xx}(f)S_{yy}(f)} \tag{10}$$

Here, $S_{xx}(f)$ and $S_{yy}(f)$ are the PSDs of $x[n]$ and $y[n]$, respectively, and $S_{xy}(f)$ is the CPSD.

These spectral densities are the Fourier transforms of the corresponding auto-correlation and cross-correlation functions. The non-negativity and boundedness of this function are a direct consequence of the Cauchy-Schwarz inequality applied to the spectral domain. Meanwhile, the diagonal elements of the coherence matrix are always unity:

$$\gamma_{ii} = \frac{|S_{ii}|^2}{S_{ii} \cdot S_{ii}} = 1 \tag{11}$$

**Theorem 1:** Softmax of band-averaged coherence is **NOT** equivalent to averaging individual softmax coherence values:

$$\text{softmax}(\bar{\gamma}_{xy}^2) \neq \frac{1}{N_b} \sum_{k \in \mathcal{B}} \text{softmax}(\gamma_{xy}^2(f_k)) \tag{12}$$

*Proof.* For two frequency bins with coherences $\gamma_1^2$ and $\gamma_2^2$, we compare Softmax of band-averaged coherence and average of individual softmax values:

$$\text{softmax}(\bar{\gamma}^2) = \frac{e^{\bar{\gamma}^2}}{\sum_j e^{\gamma_j^2}} \tag{13}$$

$$\frac{\text{softmax}(\gamma_1^2) + \text{softmax}(\gamma_2^2)}{2} = \frac{1}{2}\left(\frac{e^{\gamma_1^2}}{\sum_j e^{\gamma_j^2}} + \frac{e^{\gamma_2^2}}{\sum_j e^{\gamma_j^2}}\right) \tag{14}$$

Here,

$$\bar{\gamma}^2 = \frac{4|S_{xy,1} + S_{xy,2}|^2}{(S_{xx,1} + S_{xx,2})(S_{yy,1} + S_{yy,2})} \tag{15}$$

The inequality first arises from coherence nonlinearity: The band-averaged coherence $\bar{\gamma}^2$ is generally not equal to the arithmetic mean $\frac{\gamma_1^2 + \gamma_2^2}{2}$ due to:

$$\frac{4|S_{xy,1} + S_{xy,2}|^2}{(S_{xx,1} + S_{xx,2})(S_{yy,1} + S_{yy,2})} \neq \frac{1}{2}\left(\frac{|S_{xy,1}|^2}{S_{xx,1}S_{yy,1}} + \frac{|S_{xy,2}|^2}{S_{xx,2}S_{yy,2}}\right) \tag{16}$$

Even if $\bar{\gamma}^2 = \frac{\gamma_1^2 + \gamma_2^2}{2}$, Jensen's inequality for the convex function $e^x$ would still create inequality in most cases. Since $f(x) = e^x$ is strictly convex, Jensen's inequality states:

$$f\left(\frac{x_1 + x_2}{2}\right) < \frac{f(x_1) + f(x_2)}{2} \tag{17}$$

Equality occurs only when $\bar{\gamma}^2 = \frac{\gamma_1^2 + \gamma_2^2}{2}$ and $\gamma_1^2 = \gamma_2^2$ are both satisfied. For EEG signals, this requires:

- Identical coherences at both frequency bins.

- Specific phase and amplitude relationships in cross-spectral densities.

- Equal power spectral densities: $S_{xx,1} = S_{xx,2}$ and $S_{yy,1} = S_{yy,2}$.

Thus, the combination of coherence calculation nonlinearity and Jensen's inequality for the exponential function ensures:

$$\text{softmax}(\bar{\gamma}^2_{xy}) \neq \frac{1}{N_b} \sum_{k \in \mathcal{B}} \text{softmax}(\gamma^2_{xy}(f_k))$$

$\square$

**Theorem 2:** Under the assumption of independent frequency bins, band-averaging acts as a form of spectral smoothing that trades bias for variance reduction.

*Proof.* The variance of the arithmetic mean is:

$$\text{Var}[\bar{\gamma}^2_{xy}] = \text{Var}\left[\frac{1}{N_b} \sum_i \bar{\gamma}^2_{xy,i}\right]$$

$$= \frac{1}{N_b^2} \text{Var}\left[\sum_i \bar{\gamma}^2_{xy,i}\right]$$

$$= \frac{1}{N_b^2}\left[\sum_i \text{Var}[\bar{\gamma}^2_{xy,i}] + 2\sum_{i<j} \text{Cov}[\bar{\gamma}^2_{xy,i}, \bar{\gamma}^2_{xy,j}]\right] \tag{18}$$

If band-limited coherences are independent:

$$\text{Var}[\bar{\gamma}^2_{xy}] = \frac{1}{N_b^2} \sum_i \text{Var}[\bar{\gamma}^2_{xy,i}] = \frac{1}{N_b}\overline{\text{Var}[\bar{\gamma}^2_{xy}]} \tag{19}$$

This shows a variance reduction by factor of $\frac{1}{N_b}$ compared to individual band estimates.

$\square$

**Theorem 3:** The softmax transformation $\mathbf{L} = \text{softmax}(\bar{\gamma}^2_{xy})$ preserves ordinal relationships:

$$\bar{\gamma}^2_{xy,i} > \bar{\gamma}^2_{xy,j} \Rightarrow y_i > y_j \tag{20}$$

*Proof.* If $\bar{\gamma}^2_{xy,i} > \bar{\gamma}^2_{xy,j}$, then $e^{\bar{\gamma}^2_{xy,i}} > e^{\bar{\gamma}^2_{xy,j}}$.Since both have the same denominator:

$$\mathbf{L}_i = \frac{e^{\bar{\gamma}^2_{xy,i}}}{\sum_k e^{\bar{\gamma}^2_{xy,k}}} > \frac{e^{\bar{\gamma}^2_{xy,j}}}{\sum_k e^{\bar{\gamma}^2_{xy,k}}} = \mathbf{L}_j \tag{21}$$

$\square$

**Theorem 4:** The arithmetic mean of pseudolabels $\mathbf{L}$ satisfies standard averaging bounds:

$$\min_{i \in \{\alpha,\beta,\gamma\}} \mathbf{L}_i \leq \mathbf{L} \leq \max_{i \in \{\alpha,\beta,\gamma\}} \mathbf{L}_i \tag{22}$$

*Proof.* Let $m = \min_i(\mathbf{L}_i)$ and $M = \max_i(\mathbf{L}_i)$. Then:

$$\mathbf{L} = \frac{1}{N_b} \sum_i \mathbf{L}_i$$

$$\geq \frac{1}{N_b} \sum_i m = \frac{1}{N_b} \cdot N_b \cdot m = m$$

$$\leq \frac{1}{N_b} \sum_i M = \frac{1}{N_b} \cdot N_b \cdot M = M \tag{23}$$

Since each $\mathbf{L}_i \in [0, 1]$, we also have:

$$0 \leq \mathbf{L} \leq 1 \qquad (24)$$

$\square$

## D    EXPERIMENTAL DETAILS AND HYPERPARAMETERS

All experiments were conducted with PyTorch 2.6 (Paszke et al., 2019) on a single Nvidia GeForce RTX 3090 GPU and 16-core CPU. LLMs were used as tools to aid or polish coding and writing.

The hyperparameter choices that are the same in all data sets are shown in Table 8. The hyperparameter choices that are dataset-specific are shown in Table 9. Since the numbers of embeddings vary here, the resulting numbers of parameters also vary across datasets. The cross entropy loss function is used for STEW, the binary cross entropy loss function is used for TUAB, the weighted cross entropy loss function is used for TUEV, and the focal loss function is used for CHB-MIT. All loss function choices are consistent with previous works except for TUEV. To demonstrate the effect of the loss function choice on TUEV, a detailed examination of the choice of loss functions for TUEV is included in Fig. 4

Table 8: Default hyperparameter settings

| Parameter | pre-train | Supervised |
|---|---|---|
| Batch size | 128 | |
| Learning schedule | Cosine | |
| Optimizer | AdamW | |
| Weight decay | $1e^{-4}$ | |
| Masking ratio | 0.3 | |
| Head | 8 | |
| Learning rate | $2e^{-3}$ | $5e^{-4}$ |
| Dropout | 0.1 | 0.3 |

Table 9: Dataset-specific settings

| Parameter | STEW | TUEV | TUAB | CHB-MIT |
|---|---|---|---|---|
| Epochs | 100 | 60 | 60 | 60 |
| Patience | 20 | 5 | 5 | 5 |
| Temperature ($\tau$) | 1.2 | 1.0 | 1.0 | 1.0 |
| Sampling rate | 128 | 250 | 200 | 256 |
| Patch size | 32 | 25 | 125 | 128 |
| Embedding size | 32 | 32 | 128 | 128 |
| Learning rate (fine tune) | $5e^{-5}$ | $5e^{-4}$ | $5e^{-5}$ | $5e^{-5}$ |
| Params | 0.05M | 0.05M | 0.8M | 0.8M |

## E    EVALUATION METRICS

To evaluate the performance of our method, we employed a set of different metrics, particularly in the context of potentially imbalanced datasets. The following metrics were used for our evaluation:

- **Accuracy** is defined as the ratio of correctly predicted samples to the total number of samples, which is a straightforward and intuitive metric. However, it can be misleading in scenarios with class imbalance, where a model might achieve high accuracy by simply predicting the majority class.

- **Balanced Accuracy (B-Accuracy)** is an alternative to the standard accuracy, which can be particularly useful for imbalanced datasets. It is defined as the arithmetic mean of sensitivity (true positive rate) and specificity (false positive rate), giving equal weight to the

performance on both positive and negative classes to mitigate the bias toward the majority class.

- **AUROC** is the acronym for the area under the receiver operating characteristic (ROC) curve that measures a model's ability to distinguish between binary classes. The ROC curve plots sensitivity against specificity, which is a robust metric against class imbalance.

- **Weighted F1 Score (W-F1)** is an extension of F1 score, a harmonic mean of precision and recall. It extends F1 score by calculating the F1 score for each class independently and then averaging them, weighted by the number of true instances for each class. This ensures that the contribution of each class to the final score is proportional to its size, making it a reliable metric for datasets with multi-classes.

## F   MORE ABLATIONS

Since the accuracy difference between our band selection design and the no-band-limit setting is small (0.3%) in Table 5, we conduct an additional band selection analysis on the CHB-MIT dataset, as shown in Table 10, to further justify our band selection design. Following the same protocol as in Table 5, we report the average of linear probing and fine-tuning scores. The difference on CHB-MIT is notably larger (3.5%). Although our design does not yield significant advantage across every dataset, the fact that it achieves performance gain even without lower-frequency information in pre-training is a key finding that opens promising directions for future research.

The significant improvement observed on CHB-MIT can be explained by the presence of focal high-frequency oscillations sometimes detected on the scalp in epilepsy contexts. Learning localized spatial patterns allows the model to capture these very high-frequency, spatially confined activities, rather than broadly coherent ones observed at lower frequencies (Besio et al., 2014; Chen et al., 2021). This also helps explain why our approach outperforms foundation models on CHB-MIT. Nevertheless, further investigation is needed to better characterize the dataset conditions under which our band selection design provides significant improvements.

Table 10: Band selection analysis on average performance (%)

| Method | STEW Accuracy | CHB-MIT B-Accuracy |
|---|---|---|
| Our design | **76.1** | **86.1** |
| No band limit (All Hz) | 75.8 (↓0.3) | 82.6 (↓3.5) |

In our framework, we adopt a default decoder with attention pooling as the main configuration. Although the result with average pooling on STEW ("no attention pool") is already reported in Table 5, additional results are provided here to show that our SPR framework learns robust and efficient representations and improves supervised performance regardless of the decoder choice. We conduct further experiments using a decoder with global average pooling and a decoder with learned weighted pooling on the STEW and CHB-MIT datasets, and compare these three common decoder designs in Table 11.

For each decoder, we first report the performance obtained with purely supervised training, followed by the performance when combined with our SPR framework. The accuracy gains are 3%, 4.4%, and 5.4% on STEW, and 34.9%, 34.9%, and 32% on CHB-MIT, respectively. These results indicate that our method consistently improves each supervised baseline by a substantial margin. This ablation on the decoders clearly demonstrates the effectiveness of our SSL approach, which does not rely on the specific capacity or design of the downstream model architecture.

It is also informative to keep the encoder fixed and replace our SSL task with traditional SSL tasks. We report results for RP (Banville et al., 2021), SimCLR (Chen et al., 2020), and VICReg (Bardes et al., 2021) on STEW in Table 12. Our encoder preserves the $C$ channels rather than collapsing them into a single channel, but these SSL methods require a single-channel representation. Therefore, it is necessary to introduce an additional pooling module to reduce the $C$ channels to one. To make comprehensive comparisons, we also analyze different choices for this pooling operation.

Table 11: Performance (%) of SPR with different decoders

| Method | STEW | | CHB-MIT | |
|---|---|---|---|---|
| | Accuracy | AUROC | B-Accuracy | AUROC |
| Attention (Default) | 75.3(0.9) | 83.0(0.8) | 54.7(1.2) | 76.3(4.1) |
| **+SPR** | **78.3(1.9)** | **86.3(2.5)** | **89.6(0.8)** | **94.1(1.4)** |
| Global average | 74.3(0.4) | 82.2(0.5) | 53.9(1.5) | 76.6(2.7) |
| **+SPR** | **78.7(0.7)** | **86.6(0.4)** | **88.8(1)** | **95.1(1.1)** |
| Learned weighted | 72.4(1) | 80.1(1.1) | 55.6(1.9) | 74.8(2.7) |
| **+SPR** | **77.8(1.5)** | **85.7(1.5)** | **87.6(0.9)** | **92.9(1)** |

These results clearly show that our encoder is not intrinsically more powerful than those used in prior SSL work. The supervised method obtains 75.3% accuracy and 83.0% AUROC in Table 4, whereas all traditional SSL methods underperform this baseline. The best variant (SimCLR with average pooling) achieves 74.9% accuracy and 80.4% AUROC. When keeping the encoder fixed and simply replacing our task with other SSL tasks, performance is worse than that of the supervised baseline.

These methods degrade performance because they require a single-channel encoder design and our encoder is not suitable for their purposes. Previous SSL approaches focus on single-channel objectives, while our multi-channel SSL objective explicitly captures spatial information, which makes it essential to use a multi-channel-in, multi-channel-out encoder in our framework. However, the results show that this encoder choice alone is not the main driver of our SOTA gains.

Table 12: Performance (%) of different SSL methods on STEW

| Method | Attention | | Average | | Weighted | |
|---|---|---|---|---|---|---|
| | Accuracy | AUROC | Accuracy | AUROC | Accuracy | AUROC |
| RP | 66.5(4.4) | 72.0(6.1) | 69.1(1.9) | 75.8(2.2) | 66.9(3.3) | 72.4(3.5) |
| SimCLR | 70.1(4.6) | 75.1(5.0) | 74.9(1.1) | 80.4(2.2) | 72.0(4.1) | 77.6(3.9) |
| VICReg | 71.0(3.5) | 76.1(3.9) | 71.0(4.5) | 77.7(2.9) | 72.0(3.9) | 78.0(4.4) |
| SPR | **Accuracy: 78.3(1.9)** | | | | **AUROC: 86.3(2.5)** | |

