# OpenReview forum: "Self-Supervised Learning with Spatiality Preserving Representation for EEG Signals"
_ICLR.cc/2026/Conference — Submitted to ICLR 2026_

### Official Review · Reviewer_jRWD · 2025-10-27

**Soundness:** 2
**Presentation:** 3
**Contribution:** 3
**Rating:** 6
**Confidence:** 5

**Summary:**

This paper proposes Spatiality Preserving Representation (SPR) Learning, a self-supervised EEG framework that maintains electrode-wise spatial relationships through a novel coherence pseudolabel task. It shows promising performance improvements across multiple EEG datasets, emphasizing the role of spatial awareness in representation learning.

**Strengths:**

See the questions

**Weaknesses:**

See the questions

**Questions:**

1. While the proposed SPR framework shows strong performance in end-to-end evaluation, it remains unclear how transferable the learned representations are to other downstream decoders. To better demonstrate the utility of the learned embeddings, the authors are encouraged to verify whether SPR representations can improve classification performance when integrated with alternative decoding architectures, validating via mainstream decoding backbones.
2. The authors include several visualizations of dataset characteristics in the Appendix. However, the paper lacks interpretability analysis on the model’s internal behavior and how it aligns with key assumptions of the framework, such as whether spatially structured representations truly drive the improvements. A more thorough examination of attention maps, coherence heatmaps, or feature attribution could help validate the claimed spatial awareness and further support the design rationale.
3. The method relies on spectral domain features for pretext target generation, where frequency resolution is tied to the temporal length of EEG segments. All datasets used in this study contain relatively long segments (5–10 seconds), which favor stable frequency estimates. It remains unclear whether SPR is applicable to shorter EEG epochs, where frequency resolution may be insufficient to reliably compute coherence features in meaningful bands. Since many EEG task involve short-duration segments, the method’s effectiveness under limited temporal length should be explicitly evaluated.
4. The use of cross-power spectral density (CPSD) to characterize inter-channel dependencies is an interesting choice. However, it would be helpful for the authors to justify this choice more explicitly. Given that temporal correlation is a more direct and computationally simpler measure of channel dependency, why was it not considered or compared? Clarifying the trade-offs between CPSD and other spatial metrics would strengthen the paper’s methodological rigor.
5. The proposed use of a band-mixture approach (α, β, γ) to generate coherence pseudolabels is novel and well-motivated. However, its necessity remains unverified in the main paper. An ablation study isolating the performance of full-band vs. band-mixture pseudolabels would help determine whether mixing bands provides a meaningful benefit or introduces unnecessary complexity.
6. The authors are strongly encouraged to validate the proposed method on additional datasets with diverse tasks and recording conditions.

---

> ### Author Response · Authors · 2025-11-21
> **Rebuttal (1/2)**
>
> Thank you very much for your insightful questions. In particular, the suggestion to include a more extensive decoder analysis was very inspiring for us. We have tried our best to answer your questions and have updated the submission accordingly.
>
> **Q1: Downstream decoders.**
>
> Following your advice, we provide additional results comparing different decoder choices. In our framework, we use a default decoder with attention pooling as the main configuration. We further conduct experiments with a decoder using global average pooling and another using learned weighted pooling on the STEW and CHB-MIT datasets, and compare these three common decoder designs:
>
> | Method             | STEW Accuracy | STEW AUROC | CHB-MIT B-Accuracy | CHB-MIT AUROC |
> |--------------------|--------------|------------|---------------------|---------------|
> | Attention (Default) | 75.3(0.9)    | 83.0(0.8)  | 54.7(1.2)          | 76.3(4.1)     |
> | +SPR               | 78.3(1.9)    | 86.3(2.5)  | 89.6(0.8)          | 94.1(1.4)     |
> | Global average     | 74.3(0.4)    | 82.2(0.5)  | 53.9(1.5)          | 76.6(2.7)     |
> | +SPR               | 78.7(0.7)    | 86.6(0.4)  | 88.8(1)            | 95.1(1.1)     |
> | Learned weighted   | 72.4(1)      | 80.1(1.1)  | 55.6(1.9)          | 74.8(2.7)     |
> | +SPR               | 77.8(1.5)    | 85.7(1.5)  | 87.6(0.9)          | 92.9(1)       |
>
>
> For each decoder, we first report the performance under purely supervised training, followed by the performance when combined with our SSL method. The resulting accuracy gains are 3\%, 4.4\%, and 5.4\% on STEW, and 34.9\%, 34.9\%, and 32\% on CHB-MIT, respectively. These results show that our method consistently improves all supervised baselines by a substantial margin, and the effectiveness of our SSL approach does not depend on the specific capacity or design of the downstream architecture.
>
> **Q2: Visualizations and interpretability.**
>
> We have added more detailed explanations of the visualizations in the main text. Overall, the model exhibits very few long-range activations across distant brain regions. With horizontal and vertical lines marking anatomical boundaries, the activation maps clearly reveal predominantly local connectivity rather than long-range patterns, which is consistent with our design choice to focus on local activity via higher-frequency coherence.
>
> A concrete neuroanatomical example arises in the CHB-MIT results. When examining Fig. 7D and Fig. 8 (Appendix B), we observe structured activation patterns over functionally connected regions that are relevant for identifying focal epilepsy (Watanabe, 1989; Plummer et al., 2008; Foldvary et al., 2001). In Fig. 8, there is pronounced local activation over the **frontal** and **prefrontal** areas, as well as the **temporal** lobe in the $\beta$ and $\gamma$ bands, and these regions show correspondingly elevated local connectivity in Fig. 7D. In contrast, central and parietal regions exhibit relatively weak activation and limited local connectivity in the activation map.
>
> **Q3: EEG temporal length.**
>
> Regarding sequence length, we use 2s segments for STEW, 5s for TUEV, and 10s for TUAB and CHB-MIT. The question of how long EEG segments should be for deep learning is itself nontrivial: shorter windows can be treated as quasi-stationary, whereas longer segments may improve signal-to-noise ratio, with other trade-offs. Our conscious choice here is to follow the segment lengths used in prior SSL EEG studies (EEG2Rep and BIOT), ensuring that our results are meaningfully comparable to existing SOTA baselines.

---

> > ### Author Response · Authors · 2025-11-21
> > **Rebuttal (2/2)**
> >
> > **Q4: Temporal correlation.**
> >
> > We chose coherence instead of temporal correlation because temporal–frequency pairs have been shown to be effective in prior work such as TF-C (Zhang et al., 2022). Our pretext task naturally learns temporal–frequency dynamics by taking temporal signals as input and predicting coherence, thereby forming a temporal–frequency pairing. To further assess alternatives, we ran additional experiments using phase synchronization and temporal correlation as prediction targets and updated Table 5. Empirically, both alternatives reduce performance relative to the baseline (by **0.7** and **0.2**, respectively).
> >
> > **Q5: Full-band vs. band-mixture.**
> >
> > We have added an additional comparison between full-band and band-mixture configurations on STEW and CHB-MIT:
> >
> > | Method                 | STEW Accuracy | CHB-MIT B-Accuracy |
> > |------------------------|--------------:|-------------------:|
> > | Our design             | 76.1          | 86.1               |
> > | No band limit (All Hz) | 75.8 (↓0.3)   | 82.6 (↓3.5)        |
> >
> >
> > We report the average of linear probing and fine-tuning scores here. The performance gap on CHB-MIT is relatively large (3.5\%). Although this does not mean that our design yields significant advantages on every dataset, the fact that we achieve performance gains even after discarding lower-frequency information is a key finding and suggests promising directions for future work of spatial representation learning. To interpret the improvement on CHB-MIT, note that focal high-frequency oscillations can sometimes be observed on the scalp in epilepsy, and learning local spatial patterns allows the model to capture very high-frequency activity that is spatially confined rather than broadly coherent across the scalp (Besio et al., 2014; Chen et al., 2021). This also helps explain why our model outperforms foundation models on CHB-MIT.
> >
> > **Q6: Additional datasets and recordings.**
> >
> > We have conducted additional experiments and incorporated them into the main paper. In particular, we performed more experiments and ablations on CHB-MIT, and added a cross-domain experiment on TUAB using recordings from TUEV:
> >
> > | Dataset          | #Samples | B-Accuracy | AUROC    |
> > |------------------|---------:|-----------:|---------:|
> > | TUAB (Pre-train) | 372,510  | 81.3(0.6)  | 89.4(0.3) |
> > | + TUEV           | 425,873  | 81.7(0.2)  | 89.4(0.3) |
> >
> >
> > Here, as more pretraining samples are incorporated, B-accuracy increases and variance decreases. The results strength the claim about the effectiveness of our SSL pretext task.

---

> > > ### Author Response · Authors · 2025-11-25
> > > **References**
> > >
> > > [1] Watanabe, K. (1989). The Localization‐Related Epilepsies: Some Problems with Subclassification. Psychiatry and Clinical Neurosciences, 43(3), 471-475.
> > >
> > > [2] Plummer, C., Harvey, A. S., & Cook, M. (2008). EEG source localization in focal epilepsy: where are we now?. Epilepsia, 49(2), 201-218.
> > >
> > > [3] Foldvary, N., Klem, G., Hammel, J., Bingaman, W., Najm, I., & Luders, H. (2001). The localizing value of ictal EEG in focal epilepsy. Neurology, 57(11), 2022-2028.
> > >
> > > [4] Zhang, X., Zhao, Z., Tsiligkaridis, T., & Zitnik, M. (2022). Self-supervised contrastive pre-training for time series via time-frequency consistency. Advances in neural information processing systems, 35, 3988-4003.
> > >
> > > [5] Besio, W. G., Martínez-Juárez, I. E., Makeyev, O., Gaitanis, J. N., Blum, A. S., Fisher, R. S., & Medvedev, A. V. (2014). High-frequency oscillations recorded on the scalp of patients with epilepsy using tripolar concentric ring electrodes. IEEE journal of translational engineering in health and medicine, 2, 2000111.
> > >
> > > [6] Chen, Z., Grayden, D. B., Burkitt, A. N., Seneviratne, U., D'Souza, W. J., French, C., ... & Maturana, M. I. (2021). Spatiotemporal patterns of high-frequency activity (80–170 Hz) in long-term intracranial EEG. Neurology, 96(7), e1070-e1081.

---

### Official Review · Reviewer_uzH8 · 2025-10-29

**Soundness:** 3
**Presentation:** 3
**Contribution:** 2
**Rating:** 6
**Confidence:** 3

**Summary:**

This paper introduces Spatiality Preserving Representation (SPR) Learning, a self-supervised learning framework for EEG signal analysis that addresses a limitation in most existing methods: the loss of spatial information about electrode relationships. SPR introduces a coherence pseudo-label prediction task that teaches models to understand the topographical organization of brain signals. The method computes magnitude-squared coherence across channel pairs in specific frequency bands (alpha, beta, gamma), excluding lower frequencies that lack spatial specificity. These coherence patterns serve as pseudo-labels for the self-supervised pretext task, forcing the model to learn meaningful inter-channel spatial relationships. The authors report performance improves between 1.6 to 15.6 % across four EEG data sets (STEW, TUEV, TUAB, CHB-MIT). The idea of using spatial coherence as a pretext label is interesting and well-motivated. However, the overall novelty appears incremental, as the contribution primarily consists of introducing a new pretext task rather than a substantial methodological innovation.

**Strengths:**

- Using spatial coherence patterns as prediction targets rather than reconstruction is conceptually interesting as it directly targets brain connectivity patterns.
- The specific combination of alpha, beta and gamma bands while excluding the other bands based on spatial specificity is well-motivated.
- Strong empiral results across several EEG data sets with consistent improvements
- Compact model achieving competitive performance against much larger foundation models
- Ablations for several components (temperature, masking ratio, coherence label, model component) show quite reasonable robustness

**Weaknesses:**

- While the coherence pseudo-label seems to be novel, the overall architecture (masked autoencoding with transformers) is fairly standard.
- While justified, the exclusion of lower frequency bands and equal weighting of selected bands seems somewhat arbitrary. Adaptive or learned band selection might be more principled. Also ablation studies for a different choice of frequency bands is missing.
- While t-SNE plots show better separation, there is limited investigation into what spatial patterns the model actually learns and whether they align with known neurophysiology.
- The authors briefly mention but don't deeply address how volume conduction affects their coherence measures, which is a significant confound in EEG connectivity analysis.
- Some baseline comparisons use different preprocessing or don't use the same in-domain pre-training setup, making it harder to isolate the contribution of the coherence objective.

**Questions:**

- How do you address the fact that coherence measures in EEG are significantly affected by volume conduction, which can create spurious correlations between nearby electrodes?
- Why use equal weighting for alpha, beta and gamma bands when they have different functional significance and SNR characteristics? Have you experimented with learnable band weights or attention mechanisms over frequency bands? What happens if you include low frequency bands but with lower weights rather than excluding them entirely? Could you show ablations with different band combinations to justify your specific choice?
- Why did you use specifically coherence over other connectivity measures (e.g., phase-locking value, mutual information, Granger causality)? How would results change?
- Could you ensure all baselines use the same in-domain pre-training setup to isolate the contribution of your coherence objective?
- What spatial patterns are learned? Can you provide interpretability analyses showing which channel pairs the model focuses on?

---

> ### Author Response · Authors · 2025-11-21
> **Rebuttal (1/2)**
>
> Thank you very much for your insightful review. The question regarding volume conduction is particularly important and thought-provoking, and we appreciate you raising it. We have updated our submission in light of your reviews.
>
> **W1: Standard architecture.**
>
> One main difference between our encoder and traditional convolutional encoders such as EEG2Rep is that we avoid mixing channels, which makes explicit learning of spatial relationships feasible. In contrast, spatial representations learned by convolution-based encoders are less reliable when channel names and orders vary across datasets (for example, when channels from different brain regions are convolved together, or when the convolution does not respect anatomical correspondence). In this sense, the encoding architecture is a prerequisite for spatial representation learning in our framework. While the individual components of the encoder are not entirely new, they are combined to provide a perspective on spatial learning that has been largely overlooked in prior work. Without relying on a more powerful or complex encoder, our framework achieves better-than-SOTA performance, opening a new avenue for spatial representation learning in EEG.
>
> **W2: Band selection.**
>
> We have added an additional band selection analysis on STEW and CHB-MIT:
>
> | Method                 | STEW Accuracy | CHB-MIT B-Accuracy |
> |------------------------|--------------:|-------------------:|
> | Our design             | 76.1          | 86.1               |
> | No band limit (All Hz) | 75.8 (↓0.3)   | 82.6 (↓3.5)        |
>
>
> Here, we report the average of linear probing and fine-tuning scores. The performance gap on CHB-MIT is relatively large (3.5\%). Although this does not mean that our design provides significant advantages on every dataset, the fact that we obtain performance gains even after discarding lower-frequency information is a key finding and suggests promising directions for future work. To interpret the improvement on CHB-MIT, note that focal high-frequency oscillations can sometimes be detected on the scalp in epilepsy, and learning local spatial patterns enables the model to capture very high-frequency activity that is spatially confined rather than broadly coherent across the scalp (Besio et al., 2014; Chen et al., 2021). This also helps explain why our model outperforms foundation models on CHB-MIT.
>
> **W3: Investigation into spatial patterns.**
>
> Briefly, the model’s activation maps in Fig. 5D and Fig. 7D (Appendix B) show structured patterns that highlight functionally connected brain regions, rather than merely memorizing statistical regularities in the pretraining data. Overall, there are very few long-range activations across distant brain regions. With the aid of horizontal and vertical lines separating anatomical regions, the activation maps clearly reveal dominant local connectivity rather than long-range patterns, consistent with our design choice to focus on local activity via higher-frequency coherence.
>
> A concrete neuroanatomical example comes from the CHB-MIT results. When examining the neuroanatomy more closely using Fig. 7D and Fig. 8 (Appendix B), we observe structured activation patterns over functionally connected regions that are relevant for identifying focal epilepsy (Watanabe, 1989; Plummer et al., 2008; Foldvary et al., 2001). In Fig. 8, there is more pronounced local activation over the **frontal** and **prefrontal** areas, as well as the **temporal** lobe in the $\beta$ and $\gamma$ bands, and these areas show correspondingly elevated local connectivity in Fig. 7D. In contrast, central and parietal regions exhibit relatively weak activation and limited local connectivity in the activation map.
>
> **W4: Volume conduction.**
>
> This is an excellent point, and we address it in Q1.
>
> **W5: Baseline comparison setup.**
>
> Some prior methods (especially cross-domain foundation models) may use slightly different preprocessing pipelines, in part to handle cross-dataset issues such as noise and channel mismatches. However, most of these works explicitly state that their preprocessing is comparable to BIOT. Since our data splits and preprocessing follow BIOT, our results should also be comparable to these baselines.

---

> > ### Author Response · Authors · 2025-11-21
> > **Rebuttal (2/2)**
> >
> > **Q1: Volume conduction and nearby electrodes.**
> >
> > This is a very interesting question. After brief consultation with clinicians in the EEG community, the feedback was that they would find it problematic if there were no strong coherence between nearby electrodes. Their recommendation is not to artificially suppress adjacency effects at the modeling stage, but rather to mitigate volume conduction during preprocessing. Referencing is one common way to do so. In this paper, we follow prior pipelines and use bipolar montage referencing. With a larger number of channels, common average referencing may be an even better option.
> >
> > **Q2: Weighting for bands.**
> >
> > We did not observe any benefit from learning band weights or from applying common heuristic band weightings. Our view is that, although different cognitive tasks emphasize different bands, this information is not directly learnable in our SSL setting, since coherence is used as the target. It would be surprising if band weighting could be robustly inferred in this context. Moreover, defining task-specific band weights would require extensive additional work that could distract from the main contribution. Additional band selection analysis is included in W2.
> >
> > **Q3: Other connectivity measures.**
> >
> > To further explore other connectivity measures, we added comparisons in Table 5 using temporal correlation and phase synchronization as alternative prediction targets. Empirically, both alternatives degrade performance relative to the baseline (by **0.7** and **0.2**, respectively). More complex measures such as partial directed coherence and Granger causality are highly relevant, and we agree they could lead to interesting extensions of our framework in future work.
> >
> > **Q4: Same in-domain setup.**
> >
> > We cannot guarantee that every implementation detail is identical across all prior studies, since some SSL EEG papers do not fully disclose their preprocessing steps. However, we closely follow the BIOT preprocessing pipeline, which provides a standard protocol, and both cross-domain and in-domain methods generally claim to be comparable to BIOT. As discussed in W5, our setup should therefore be comparable to previous baselines, with full details provided in Appendix A.
> >
> > **Q5: Spatial pattern interpretability.**
> >
> > This question is answered in W3.

---

> > > ### Author Response · Authors · 2025-11-25
> > > **References**
> > >
> > > [1] Besio, W. G., Martínez-Juárez, I. E., Makeyev, O., Gaitanis, J. N., Blum, A. S., Fisher, R. S., & Medvedev, A. V. (2014). High-frequency oscillations recorded on the scalp of patients with epilepsy using tripolar concentric ring electrodes. IEEE journal of translational engineering in health and medicine, 2, 2000111.
> > >
> > > [2] Chen, Z., Grayden, D. B., Burkitt, A. N., Seneviratne, U., D'Souza, W. J., French, C., ... & Maturana, M. I. (2021). Spatiotemporal patterns of high-frequency activity (80–170 Hz) in long-term intracranial EEG. Neurology, 96(7), e1070-e1081.
> > >
> > > [3] Watanabe, K. (1989). The Localization‐Related Epilepsies: Some Problems with Subclassification. Psychiatry and Clinical Neurosciences, 43(3), 471-475.
> > >
> > > [4] Plummer, C., Harvey, A. S., & Cook, M. (2008). EEG source localization in focal epilepsy: where are we now?. Epilepsia, 49(2), 201-218.
> > >
> > > [5] Foldvary, N., Klem, G., Hammel, J., Bingaman, W., Najm, I., & Luders, H. (2001). The localizing value of ictal EEG in focal epilepsy. Neurology, 57(11), 2022-2028.

---

### Official Review · Reviewer_ePRy · 2025-10-31

**Soundness:** 1
**Presentation:** 2
**Contribution:** 1
**Rating:** 2
**Confidence:** 4

**Summary:**

The paper introduces a self-supervised learning framework for EEG called Spatiality Preserving Representation (SPR) learning. The authors contribute a novel pretext task in which the model aims to predict a band-mixture coherence matrix, which encodes the connectivity between electrode pairs. The authors provide evaluations of their methodology on multiple datasets and it's claimed to achieve new state-of-the-art performance on multiple EEG benchmarks with in some cases showing massive gains (e.g. CHB-MIT).

**Strengths:**

1. The pretraining task explores a novel and promising direction, which is welcome to the field, as it does not focus on input-based reconstruction.
2. The paper is generally easy to read and the methodology is clearly explained.
3. The authors perform evaluation on multiple datasets and provide ablation results.

**Weaknesses:**

1. **Fundamental concerns with experimental validation**: The paper's central claim is that its pretext task leads to superior representations. However, the results appear to contradict this. Specifically, the SPR (**Random**) model, which is just the paper's transformer model trained from scratch without pretraining, achieves higher performance than the baseline methods on nearly every metric. For example, on TUEV, SPR (Random) scores 58.1% compared to the their best pretrained comparison method EEG2Rep at 53%. SPR (Random) is furthermore 12-17% better than all other methods.
This implies that the model architecture is the dominant driver of performance, not the novel pretext task. However, no ablations or explanations are provided for this. Given that there doesn't appear to be a lot of novelty in the encoder (a single-channel encoder isn't new in and of itself, for example Mohsenvand et al. (2020) used one (which the authors cite but do not compare against), I am concerned that either A) the models are not evaluated under strict comparable conditions (such as dataset versions and splits) or B) the baseline models were poorly evaluated.

2. **Implausibility of the pretext task**: The pretext task itself seems too simple to be the source of such large performance gains. The model is only asked to predict a single scalar value per channel pair. I don't understand how learning this extremely low-resolution spatial map could produce such powerful, state-of-the-art representations, especially when compared to the complexity of reconstruction or contrastive tasks. The fact that the SPR(Random) model performs so well I think reinforces this skepticism: the novel pretext task does not appear to be the critical ingredient. These concerns I think are quite self-evident from the paper in its current state I would think and should be addressed.

3. **Inconsistent comparisons**:
In the main results, SPR is compared against BIOT, which is presented by the original authors as a cross-domain model. However, more recent methods such as Labram and Cbramod, from a similar cross-domain paradigm but which sometimes outperform SPR, are placed in the appendix instead.

4. **Strange results**: The single most impressive result to me, which is a 15.6% balanced accuracy improvement on CHB-MIT (89.6% versus 74.0% for the next best model) is only presented in the appendix. A performance gain of this size should be a central finding of the paper, I would think. The fact its not included in the main paper I find puzzling and concerning. Naturally, here too I remain unconvinced whether a strictly identical evaluation protocol was used for SPR and comparison methods. If there was, it would be simple to run ablation experiments that would be extremely valuable to the EEG community to explain where these massive performance gains come from (is it here too just the architecture?).

5.  **Lack of justification for methodological choices**:
The authors surprisingly exclude delta and theta bands. Their argumentation is plausible, but these bands are widely deemed *critical* for many EEG tasks, including seizure detection (!). Their strong claim that these bands should be omitted requires empirical validation, which is lacking.

6. The ablation in Table 5 shows that a single wide band (8-100Hz) results only in a drop of 1.3% performance, suggesting that the author's complex band-mixture approach provides only a marginal benefit. This raises the question how the model is able to learn the highly useful features during pretraining that give it such strong downstream performance.



Minor: The comparisons lack some earlier EEG-specific methods which do not rely on transformer architectures or reconstruction. I realize one cannot compare exhaustively, but a lot of the initial EEG literature seems missing. Just as an example, I think it would be valuable if methods like Banville et al. (2021; 10.1088/1741-2552/abca18) would be included.

**Questions:**

1. The SPR(Random) model outperforms nearly all SOTA SSL baselines. Can you please explain this? What specific architectural component is responsible?

2. Given that the pretext task is to predict a single (or 3-avg) scalar per channel pair, how do you explain this extremely low-resolution task leading to such powerful representations? This seems implausible, especially when your random baseline is already so strong.

3. Why are comparisons to major foundation models (LaBraM, CBraMod) and the CHB-MIT results (Table 6) placed in the appendix , while the (less relevant) cross-domain BIOT model is in the main results?

4. Given the 15.6% accuracy improvement on CHB-MIT, why was this not a central result? Can you provide more analysis, including a comparison to your SPR(Random) architecture trained from scratch on this dataset?

5. Can you provide an empirical ablation showing the performance impact of including the δ (0.5-4 Hz) and θ (4-8 Hz) bands in your pretext task?

6. The performance drop for using a single wide band (8-100 Hz) vs. your "band mixture" is only 1.3% (Table 5). This suggests the band-mixture component offers minimal benefit, but more concerningly, it indicates that a extremely low-resolution pretraining task is almost as good as your complete method. Could you comment on this?

---

> ### Author Response · Authors · 2025-11-21
> **Rebuttal (1/4)**
>
> Thank you very much for your detailed review. We are grateful for your constructive criticisms and have worked hard to improve the quality of the paper based on your feedback. We have also updated the submission accordingly.
>
> **W1: Concerns about the performance of the SPR (Random) model.**
>
> The data splits and preprocessing strictly follow prior work such as EEG2Rep and BIOT, with full details provided in Appendix A. For STEW and TUEV, our supervised model indeed outperforms previous SSL methods (Table 4), which can be partly explained by the fact that convolution-based encoders in earlier work did not account for channel mismatches across datasets, leading to suboptimal performance when channel names or orders differ. In contrast, our architecture is inherently insensitive to channel naming and ordering. It is also important to note that our supervised model does not surpass BIOT, TS-TCC, or EEG2Rep on TUAB in Table 4, whereas our SSL model outperforms all methods on TUAB. In our view, the 3\%, 4.6\%, and 2.9\% accuracy improvements of our SSL model over our supervised model on STEW, TUEV, and TUAB, respectively, already provide strong evidence in favor of the proposed SSL pretext task.
>
> We also acknowledge that the architecture itself may contribute to the stronger supervised performance in Table 4. However, our model is substantially less complex than recent foundation models, so it should be clear that architectural gains alone do not explain why our SSL approach achieves comparable or superior results relative to these large models. To clarify this point, we additionally replace our attention-pooling decoder with other mainstream decoders and compare supervised versus SSL performance, showing that the benefits of our SSL framework are not tied to a specific decoder choice:
>
> | Method             | STEW Accuracy | STEW AUROC | CHB-MIT B-Accuracy | CHB-MIT AUROC |
> |--------------------|--------------|------------|---------------------|---------------|
> | Attention (Default) | 75.3(0.9)    | 83.0(0.8)  | 54.7(1.2)          | 76.3(4.1)     |
> | +SPR               | 78.3(1.9)    | 86.3(2.5)  | 89.6(0.8)          | 94.1(1.4)     |
> | Global average     | 74.3(0.4)    | 82.2(0.5)  | 53.9(1.5)          | 76.6(2.7)     |
> | +SPR               | 78.7(0.7)    | 86.6(0.4)  | 88.8(1)            | 95.1(1.1)     |
> | Learned weighted   | 72.4(1)      | 80.1(1.1)  | 55.6(1.9)          | 74.8(2.7)     |
> | +SPR               | 77.8(1.5)    | 85.7(1.5)  | 87.6(0.9)          | 92.9(1)       |
>
> For each decoder, we first report the purely supervised performance, followed by the performance when combined with our SSL method. The corresponding accuracy gains are 3\%, 4.4\%, and 5.4\% on STEW, and 34.9\%, 34.9\%, and 32\% on CHB-MIT, respectively, showing that our method consistently improves all supervised baselines by a substantial margin.
>
> Regarding the large performance gains on TUEV for our supervised model, these are closely related to the choice of loss function. For a highly imbalanced dataset like TUEV, it is difficult to justify not using a weighted loss in supervised learning. However, the results with weighted cross-entropy (W-CE) were strong, which raised concerns for us as well. In our initial submission, we have already connected this observation to Fig. 4 in the ablation study. To more clearly illustrate the impact of loss functions, we also move the results from Fig. 4 to the table here (noting that we do not know which loss function EEG2Rep used for TUEV):
>
> | Method          | B-Accuracy W-CE loss | B-Accuracy CE loss | B-Accuracy Focal loss | W-F1 W-CE loss | W-F1 CE loss | W-F1 Focal loss |
> |----------------|----------------------|--------------------|------------------------|----------------|--------------|-----------------|
> | EEG2Rep        | 53(1.6)              | 53(1.6)            | 53(1.6)                | 75.1(1.2)      | 75.1(1.2)    | 75.1(1.2)       |
> | SPR (Random)   | 58.1(2.1)            | 46.2(4.9)          | 44.4(5.5)              | 75.5(2.7)      | 77.1(1.4)    | 76.6(2.5)       |
> | **SPR (Pre-train)** | **62.7(1.7)**       | 55.1(2.7)          | 53.5(2.6)              | 78.1(2.4)      | **82.5(0.9)**| **82.5(1.8)**   |
>
>
> If we instead consider the cross-entropy (CE) loss results, our supervised model no longer outperforms EEG2Rep, while our SSL model achieves a 2.1\% improvement in B-accuracy and a 7.4\% improvement in W-F1. In this view, the gain in B-accuracy is less striking, but W-F1 improves, indicating a trade-off between the two metrics. Nevertheless, the table still supports a consistent advantage of our SSL method irrespective of the loss function choice. We have added our loss-function settings for other datasets, which are aligned with prior work, in Appendix D.

---

> > ### Author Response · Authors · 2025-11-21
> > **Rebuttal (2/4)**
> >
> > **W2: Understanding the pretext task in self-supervised learning.**
> >
> > A key difference between our encoder and traditional convolutional encoders such as EEG2Rep is that we avoid mixing channels, which makes explicit learning of spatial relationships feasible. In contrast, spatial representations learned by convolution-based encoders are less reliable when channel names and orders vary across datasets (e.g., whether channels from different brain regions are convolved together, and whether the convolution respects anatomical correspondence). In this sense, the encoding architecture is a prerequisite for our spatial representation learning. At the same time, the complexity of a pretext task does not necessarily reflect its novelty. The effectiveness of a fixed coherence target is closely related to masked modeling: masks introduce variability and randomness that act as augmentations of the original samples. SimCLR has argued that appropriate augmentations are the most critical component for SSL, and Table 5 likewise shows that suitable masking hyperparameters are crucial for our framework to function well. Importantly, this dependence on proper masking is ubiquitous across relevant SSL work in both vision and time-series domains and is not a specific weakness of our framework. Overall, our framework differs substantially from current SOTA methods and achieves better-than-SOTA performance, opening a new avenue for spatial representation learning in EEG.
> >
> > Our pretext task also encourages learning of temporal–frequency dynamics because the model receives temporal signals as input and coherence as the prediction target, forming temporal–frequency pairs. Such pairs have been shown to be effective in prior work like TF-C (Zhang et al., 2022). In addition, we conducted new experiments with alternative prediction targets, including phase synchronization and temporal correlation, and updated Table 5 accordingly. Empirically, these alternatives degrade performance relative to the baseline (by **0.7** and **0.2**, respectively).
> >
> > **W3: Cross-domain paradigm.**
> >
> > We apologize for any confusion caused by mixed terminology in this area. In the context of this paper, the distinction between in-domain and cross-domain pretraining is whether the model is pretrained only on a pretraining subset of a single dataset (in-domain) or jointly on multiple datasets (cross-domain). Under this definition, our method belongs to the in-domain category, which is why we did not include comparisons with foundation models such as LaBraM and CBraMod in the main text of the initial submission. We also note that BIOT appears in both in-domain and cross-domain result tables under different experimental setups.
> >
> > **W4: Presentation of the CHB-MIT results.**
> >
> > We agree that the CHB-MIT results are striking and should be highlighted as a central contribution of the paper. As clarified in W3, our work is in-domain, whereas CHB-MIT has previously been used mainly for cross-domain pretraining comparisons rather than in-domain ones, which is why we initially placed the CHB-MIT analysis in the appendix. We apologize for this underemphasis in the original submission. Because we strictly follow the BIOT preprocessing pipeline, our results are directly comparable to prior methods. With the additional page allowed in the discussion phase, we conducted further experiments and moved the foundation-model comparison into the main text (Tables 6 and 7). The SPR (Random) results are now reported in Table 6, with 54.7\% B-accuracy and 76.3\% AUROC, and more ablations on CHB-MIT demonstrating SSL improvements over the supervised baseline are provided in the reply to W1 via the decoder performance table.

---

> > > ### Author Response · Authors · 2025-11-21
> > > **Rebuttal (3/4)**
> > >
> > > **W5: Excluding lower-frequency bands.**
> > >
> > > The concern about discarding lower-frequency bands is well founded, as these bands carry important cognitive information and should not be removed in supervised training. In our framework, band selection is applied only during pretraining for the spatial-prediction task. For downstream medical label prediction, the model is still trained via linear probing or finetuning on the full set of frequency bands, without any band restriction.
> > >
> > > Figures 6 and 8 in Appendix B show that higher-frequency bands exhibit stronger spatial specificity, whereas spatial information in lower-frequency bands appears more diffuse. This higher-frequency spatial specificity has also been widely reported in the neuroscience literature (Ł˛eski et al., 2013;
> > > Arnulfo et al., 2020; Han et al., 2021). Lower-frequency activity tends to spread over large scalp areas, while high-frequency components attenuate more rapidly with distance. Coherence decreases with distance, and higher frequencies show steeper falloffs, indicating that high-frequency activity is more local and low-frequency activity supports longer-range interactions. By constraining learning to higher-frequency bands during pretraining, the model is encouraged to capture the locality of channels. Figures 6 and 8 further confirm that our model indeed learns predominantly local rather than long-range spatial relationships, as channel pairs within nearby brain regions are much more strongly activated.
> > >
> > > **W6: Single wide-band performance.**
> > >
> > > In Table 5, the 1.3\% drop in performance when using a single band is meaningful, given that the overall gain from SSL over the supervised baseline is 3\%. However, the 0.3\% drop with no band selection is understandably less convincing on its own. We fully recognize that analyzing band selection on STEW alone is insufficient to fully justify the design, so we have added an additional comparative analysis on CHB-MIT to strengthen this point:
> > >
> > > | Method                 | STEW Accuracy | CHB-MIT B-Accuracy |
> > > |------------------------|--------------:|-------------------:|
> > > | Our design             | 76.1          | 86.1               |
> > > | No band limit (All Hz) | 75.8 (↓0.3)   | 82.6 (↓3.5)        |
> > >
> > >
> > > We report the average of linear probing and fine-tuning scores. The performance gap on CHB-MIT is relatively large (3.5\%). Although this does not imply that our design yields significant advantages on every dataset, the fact that we obtain performance gains even after discarding lower-frequency information is a key finding and points to promising directions for future work. To interpret the improvement on CHB-MIT, we note that focal high-frequency oscillations can sometimes be observed on the scalp in epilepsy, and learning local spatial patterns enables the model to capture very high-frequency activity that is spatially confined rather than broadly coherent across the scalp (Besio et al., 2014; Chen et al., 2021). This also helps explain why our model outperforms foundation models on CHB-MIT.
> > >
> > > **W7: Earlier EEG-specific methods.**
> > >
> > > We appreciate the reviewer’s suggestion and have carefully read Banville et al. (2021). However, their work adopts a preprocessing pipeline that differs substantially from the one used in our line of research, making direct comparisons difficult. Due to this mismatch, it is challenging to provide a fair quantitative comparison. We added additional related literature into the introduction section.

---

> > > > ### Author Response · Authors · 2025-11-21
> > > > **Rebuttal (4/4)**
> > > >
> > > > **Q1: SPR (Random) model performance.**
> > > >
> > > > This point is addressed in W1. To elaborate briefly, we also present the new cross-domain pretraining results from Table 7 here:
> > > >
> > > > | Dataset          | #Samples | B-Accuracy | AUROC    |
> > > > |------------------|---------:|-----------:|---------:|
> > > > | TUAB (Pre-train) | 372,510  | 81.3(0.6)  | 89.4(0.3) |
> > > > | + TUEV           | 425,873  | 81.7(0.2)  | 89.4(0.3) |
> > > >
> > > >
> > > > Here, the cross-domain pretraining results also address the concern that SPR performs well merely because of a strong base model. As more pretraining samples are incorporated, B-accuracy increases and variance decreases, clearly demonstrating the effectiveness of the coherence-based prediction pretext task.
> > > >
> > > > **Q2: Spatial resolution.**
> > > >
> > > > The aspects related to the SSL logic are answered in W2. Regarding spatial resolution, current SOTA SSL EEG studies rarely use more than about 20 channels in their preprocessed datasets. Artificially increasing spatial resolution when the original recordings have low spatial density is itself problematic. For this reason, we follow the preprocessing protocols of prior work such as EEG2Rep and BIOT, which use 14 channels for STEW and 16 channels for TUEV, TUAB, and CHB-MIT, ensuring that our results are meaningfully comparable to existing SOTA baselines.
> > > >
> > > > Given that our framework already performs well under low spatial resolution, it is reasonable to expect even greater benefits when applied to higher-resolution recordings, which is a relatively underexplored regime in current SSL EEG research. We have released our code in the supplementary material and will provide both code and pretrained weights upon publication. We have made every effort to ensure reproducibility and are committed to further supporting the EEG community in using our code and pretrained models.
> > > >
> > > >
> > > > **Q3 and Q4: Placement of the CHB-MIT results.**
> > > >
> > > > The questions are answered in W3 and W4. While the CHB-MIT performance is particularly strong, we observe structured activation patterns that highlight functionally connected brain regions relevant for identifying focal epilepsy (Watanabe, 1989; Plummer et al., 2008; Foldvary et al., 2001). In Fig. 8, there is more pronounced local activation over the **frontal** and **prefrontal** areas, as well as the **temporal** lobe in the $\beta$ and $\gamma$ bands, with these regions showing correspondingly elevated local connectivity in Fig. 7D. In contrast, central and parietal regions exhibit relatively weak activation and limited local connectivity in the learned heat map. Overall, the map reveals predominantly local rather than long-range connectivity, aligning with our design choice to emphasize local connectivity via higher-frequency coherence.
> > > >
> > > > **Q5 and Q6: Including lower-frequency bands.**
> > > >
> > > > The questions are answered in W6.

---

> > > > > ### Author Response · Authors · 2025-11-25
> > > > > **References**
> > > > >
> > > > > [1] Zhang, X., Zhao, Z., Tsiligkaridis, T., & Zitnik, M. (2022). Self-supervised contrastive pre-training for time series via time-frequency consistency. Advances in neural information processing systems, 35, 3988-4003.
> > > > >
> > > > > [2] Łęski, S., Lindén, H., Tetzlaff, T., Pettersen, K. H., & Einevoll, G. T. (2013). Frequency dependence of signal power and spatial reach of the local field potential. PLoS computational biology, 9(7), e1003137.
> > > > >
> > > > > [3] Arnulfo, G., Wang, S. H., Myrov, V., Toselli, B., Hirvonen, J., Fato, M. M., ... & Palva, J. M. (2020). Long-range phase synchronization of high-frequency oscillations in human cortex. Nature communications, 11(1), 5363.
> > > > >
> > > > > [4] Han, C., Wang, T., Yang, Y., Wu, Y., Li, Y., Dai, W., ... & Xing, D. (2021). Multiple gamma rhythms carry distinct spatial frequency information in primary visual cortex. PLoS biology, 19(12), e3001466.
> > > > >
> > > > > [5] Besio, W. G., Martínez-Juárez, I. E., Makeyev, O., Gaitanis, J. N., Blum, A. S., Fisher, R. S., & Medvedev, A. V. (2014). High-frequency oscillations recorded on the scalp of patients with epilepsy using tripolar concentric ring electrodes. IEEE journal of translational engineering in health and medicine, 2, 2000111.
> > > > >
> > > > > [6] Chen, Z., Grayden, D. B., Burkitt, A. N., Seneviratne, U., D'Souza, W. J., French, C., ... & Maturana, M. I. (2021). Spatiotemporal patterns of high-frequency activity (80–170 Hz) in long-term intracranial EEG. Neurology, 96(7), e1070-e1081.
> > > > >
> > > > > [7] Banville, H., Chehab, O., Hyvärinen, A., Engemann, D. A., & Gramfort, A. (2021). Uncovering the structure of clinical EEG signals with self-supervised learning. Journal of Neural Engineering, 18(4), 046020.
> > > > >
> > > > > [8] Watanabe, K. (1989). The Localization‐Related Epilepsies: Some Problems with Subclassification. Psychiatry and Clinical Neurosciences, 43(3), 471-475.
> > > > >
> > > > > [9] Plummer, C., Harvey, A. S., & Cook, M. (2008). EEG source localization in focal epilepsy: where are we now?. Epilepsia, 49(2), 201-218.
> > > > >
> > > > > [10] Foldvary, N., Klem, G., Hammel, J., Bingaman, W., Najm, I., & Luders, H. (2001). The localizing value of ictal EEG in focal epilepsy. Neurology, 57(11), 2022-2028.

---

> > > > > > ### Comment · Reviewer_ePRy · 2025-11-25
> > > > > >
> > > > > > Thank you for the detailed rebuttal and additional experiments. These certainly clarify several points, but some central concerns remain about the evaluation design and about how to interpret the claimed contributions.
> > > > > >
> > > > > > With the new evaluations, it is clear that your SSL pretraining method improves downstream performance. This is interesting, but in and of itself, not yet useful for the community. We by now know of many (diverse) strategies which can improve performance versus randomly initialized models, especially for transformers, which lack sensible time series inductive biases. I still find it very difficult to know whether your result is relevant or not, which I attribute to the evaluation scheme. Some further questions:
> > > > > >
> > > > > > **1.** Can you elaborate on the explanation that prior baselines underperform due to channel mismatches? For example, BIOT's evaluation standardizes TUEV channels, where do the channel mismathces come in? Are you making a statement about the (multi-dataset) pretraining of the methods you compare to? Because from what I can tell, in Table 3+4 you compare against cross-domain transformer models. But you're proposing an in-domain method here. Why are you comparing to a different type of approach? If cross-domain methods transfer poorly to the datasets you're evaluating due to channel mismatches, how do you know your method is SOTA? I
> > > > > >
> > > > > > **2.** If you choose to rely on cross-domain comparisons, they need to be complete. Methods like LaBraM and CBraMod appear for some datasets but are missing for TUEV and STEW. This inconsistency makes it difficult to interpret the SOTA claim. On what basis is that claim made if the strongest available cross-domain models are not uniformly included? (I apologize if I have misunderstood which methods are within and cross domain, but I don't think this is currently conveyed to the reader.)
> > > > > >
> > > > > > **3.** From what I can tell, you actually do include within-domain models, but these are included in the "Foundation model" table? Why? Are models like EEGNet, SparcNet, EEGConformer pretrained in some way?
> > > > > >
> > > > > > To summarize:
> > > > > >
> > > > > > You demonstrate that your SSL pretraining helps relative to random initialization. To understand its practical relevance, the comparison needs to include other methods trained on exactly the same data and under the same assumptions. The unresolved question is whether your approach improves over existing within-domain pretraining strategies. As a reader, I still cannot answer that.
> > > > > >
> > > > > > You don't need Banville et al. methods for this. But using such a method, or any other SSL method, with your encoder and data, would be much more informative than many of the provided evaluations.

---

> ### Author Response · Authors · 2025-11-26
> **Thank you very much for your thoughtful comments.**
>
> Dear Reviewer ePRy,
>
> We truly appreciate your thoughtful comments and would like to address several of your concerns that can be clarified quickly, and we plan to add further experiments for Points 2 and 3, which will require more time.
>
> **Channel mismatches.**
>
> Consider two datasets, A and B. There is no guarantee that dataset B uses exactly the same channel names as A, or that authors always reorder channels to ensure consistent alignment, even when names match. By channel mismatches, we refer to a situation that is particularly problematic for **in-domain methods**: these methods are typically **developed**, pretrained, and evaluated on dataset A, and then a new model is trained from **scratch** on dataset B (pretrain and test on B, report results on B). When channels are processed via convolutions, performance on B can be suboptimal compared to A because the model architecture implicitly assumes a specific channel alignment that may not hold for B.
>
> In our case, although our method is in-domain, our architecture does not perform convolutions across channels, so mismatches in channel naming or ordering do not pose the same issue.
>
> Cross-domain foundation models such as CBraMod and LaBraM typically adopt explicit channel-alignment strategies, because they are **pretrained jointly on samples from both A and B**, and thus must enforce a consistent channel layout at the first place.
>
> **In-domain/cross-domain pretraining and SOTA choice.**
>
> When we refer to our method as in-domain, we emphasize that we do not design any special mechanism to pretrain a single model simultaneously on **samples from both datasets A and B**. Instead, our reported results follow the standard in-domain protocol: pretrain and evaluate on dataset A, then independently pretrain a new model from scratch on dataset B and evaluate on B. Under this setting, EEG2Rep is the primary SOTA baseline we consider.
>
> In contrast, cross-domain foundation models first pretrain a single model on combined samples from A and B, then finetune that pretrained model on A (report on A) and B (report on B). Their performance naturally benefits from the larger amount of pretraining samples.
>
> It is an excellent point that Table 6 compares LaBraM and CBraMod to some in-domain methods, while our main results in Table 4 do not include LaBraM and CBraMod. Our initial intention was to treat EEG2Rep as the main SOTA in the in-domain setting and to present LaBraM and CBraMod as complementary cross-domain baselines. Thus, our table is presented this way to best illustrate the results that have already been reported in the original papers. **Inside each table, the methods are comparable.** Meanwhile, we agree that our presentation would be more consistent if we also fine-tune LaBraM and CBraMod using their public code and pretrained weights and report these results directly in Table 4. We plan to run these additional experiments and will report the new results as soon as they are available.

---

> > ### Comment · Reviewer_ePRy · 2025-11-27
> >
> > Dear Authors,
> >
> > Thank you for the preliminary response. I appreciate your commitment to fixing the inconsistency in the cross-domain comparisons (Point 2) by running LaBraM/CBraMod on the missing datasets.
> >
> > However, I want to be clear about what to me would be necessary to recommend the paper for acceptance, so that you do not spend your limited rebuttal time on experiments that do not address the core concerns.
> >
> > 1. A "Control" Experiment is critical (Ref: Point 2 & Summary) While adding LaBraM results is helpful, it does not isolate your contribution. To prove your SSL method is the source of the performance gain (rather than your specific encoder architecture), what is necessary is a comparison of [Your Encoder + Standard SSL (e.g., Banville/SimCLR/...)] vs. [Your Encoder + Your Proposed SSL]. Without this ablation, it is impossible to determine if your results are due to the pretraining strategy, simply a better backbone, or any remaining miscellaneous factors such as channel mismatches.
> >
> > 2. Clarification on "Channel Mismatch" (Ref: Point 1) Your explanation regarding channel mismatches remains unclear to me for the in-domain setting. If a baseline model is trained on TUEV and tested on TUEV, the channel layout is identical. Please clarify why you believe "channel mismatch" is a valid explanation for baseline underperformance in strict in-domain evaluations.

---

> ### Author Response · Authors · 2025-12-02
> **Thank you very much for making explicit requirement**
>
> We thank the reviewer for pointing out what is necessary for acceptance here. We have tried our best to address the two remaining concerns and updated the submission accordingly.
>
> **Fixed encoder and previous SSL methods**
>
> We agree that it is informative to keep the encoder fixed and replace our SSL task with traditional SSL tasks. We report results for RP (Banville et al., 2021), SimCLR (Chen et al., 2020), and VICReg (Bardes et al., 2021) on STEW.
>
> Our encoder preserves the $C$ channels rather than collapsing them into a single channel, but these SSL methods require a single-channel representation. Therefore, it is necessary to introduce an additional pooling module to reduce the $C$ channels to one. To make comprehensive comparisons, we also analyze different choices (attention, average, and weighted poolings) for this pooling operation. We have updated Appendix F in the paper and also present the table here:
>
> | Method | Attention Accuracy | Attention AUROC | Average Accuracy | Average AUROC | Weighted Accuracy | Weighted AUROC |
> |--------|--------------------|-----------------|------------------|---------------|-------------------|----------------|
> | RP     | 66.5(4.4)          | 72.0(6.1)       | 69.1(1.9)        | 75.8(2.2)     | 66.9(3.3)         | 72.4(3.5)      |
> | SimCLR | 70.1(4.6)          | 75.1(5.0)       | 74.9(1.1)        | 80.4(2.2)     | 72.0(4.1)         | 77.6(3.9)      |
> | VICReg | 71.0(3.5)          | 76.1(3.9)       | 71.0(4.5)        | 77.7(2.9)     | 72.0(3.9)         | 78.0(4.4)      |
>
> These results clearly show that our encoder is not intrinsically more powerful than those used in prior SSL work. The supervised method obtains 75.3\% accuracy and 83.0\% AUROC in Table 4, whereas all traditional SSL methods underperform this baseline. The best variant (SimCLR with average pooling) achieves 74.9\% accuracy and 80.4\% AUROC. When keeping the encoder fixed and replacing our task with other SSL tasks, performance is worse than that of the supervised baseline.
>
> These methods degrade performance because they require a single-channel encoder design and our encoder is not suitable for their purposes. Previous SSL approaches focus on single-channel objectives, while our multi-channel SSL objective explicitly captures spatial information, which makes it essential to use a multi-channel-in, multi-channel-out encoder in our framework. However, the results above show that this encoder choice alone is not the main driver of our SOTA results (78.3 \% accuracy and 86.3\% AUROC on STEW).
>
> **Channel order mismatches**
>
> We would like to clarify that channel orders do not matter for in-domain methods whose encoder immediately collapses all input channels into a single channel representation. In this case, no explicit cross-channel information is modeled, and the output is also single-channel. However, channel-order mismatches can become problematic when the encoder applies spatially structured cross-channel operations (e.g., grouping channels by anatomical region) or when channels are tied to predefined probability/weight distributions or spatial encodings.
>
> For instance, CNN-Transformer (Peh et al., 2022) use segment-level classifiers with probabilities defined over seven regions: frontal, frontal-temporal, non-frontal (all non-frontal channels), central, parietal, occipital, and the entire scalp. Another problem is the use of regional convolutions or regional attention mechanisms. For example, Cui et al., 2020 capture regional information among adjacent electrodes by applying convolutional layers with fixed kernel sizes to learn regional patterns. Due to channel order mismatches, such frameworks designed on the STEW dataset need adjustments when applied to TUEV.
>
>
> **References**
>
> [1] Banville, H., Chehab, O., Hyvärinen, A., Engemann, D. A., & Gramfort, A. (2021). Uncovering the structure of clinical EEG signals with self-supervised learning. Journal of Neural Engineering, 18(4), 046020.
>
> [2] Chen, T., Kornblith, S., Norouzi, M., & Hinton, G. (2020, November). A simple framework for contrastive learning of visual representations. In International conference on machine learning (pp. 1597-1607). PmLR.
>
> [3] Bardes, A., Ponce, J., & LeCun, Y. (2021). Vicreg: Variance-invariance-covariance regularization for self-supervised learning. arXiv preprint arXiv:2105.04906.
>
> [4] Peh, W. Y., Yao, Y., & Dauwels, J. (2022, July). Transformer convolutional neural networks for automated artifact detection in scalp EEG. In 2022 44th Annual International Conference of the IEEE Engineering in Medicine & Biology Society (EMBC) (pp. 3599-3602). IEEE.
>
> [5] Cui, H., Liu, A., Zhang, X., Chen, X., Wang, K., & Chen, X. (2020). EEG-based emotion recognition using an end-to-end regional-asymmetric convolutional neural network. Knowledge-Based Systems, 205, 106243.

---

### Official Review · Reviewer_4ps2 · 2025-11-01

**Soundness:** 2
**Presentation:** 2
**Contribution:** 2
**Rating:** 4
**Confidence:** 4

**Summary:**

This paper proposes a self-supervised Spatiality Preserving Representation (SPR) framework that pre-trains by predicting band-limited channel–channel coherence to capture spatial relationships. It achieves consistent state-of-the-art performance on four benchmark datasets under both linear probing and fine-tuning.

**Strengths:**

S1. SPR introduces a novel SSL framework that preserves spatial electrode topology, bridging functional connectivity analysis with EEG representation learning.

S2. The band-mixture coherence pretext task efficiently models spatial structures.

S3. The method consistently outperforms prior EEG SSL approaches across multiple datasets, demonstrating strong generalization and robustness.

**Weaknesses:**

W1. The coherence-based pseudolabel measures only the average linear synchronization between channels in the frequency domain. While simple, it is essentially a low-order statistic that cannot capture directional, nonlinear, or dynamic interactions. Moreover, the spatial patterns may be partially biased by volume conduction, as also noted in Appendix B.

W2. Temporal modeling appears relatively weak. Table 5 suggests that the framework relies heavily on positional encoding, indicating that it primarily learns static spatial structures rather than time-evolving EEG dynamics.

W3. The band-mixture strategy may obscure band-specific physiological characteristics, particularly in cognitive or disease-related tasks where frequency-band specificity is crucial.

W4. Although the experiments cover four datasets (STEW, TUAB, TUEV, CHB-MIT), they share similar setups, including low-density, short-segment EEG with comparable sampling rates and channel counts. Such homogeneity limits the generalizability of claims about broad applicability. It would be valuable to test the method on more diverse paradigms such as MI, SSVEP, or neurodegenerative-disease datasets to better demonstrate the advantages of spatial coherence modeling.

**Questions:**

Q1. Could the authors elaborate on how the learned spatial representations correspond to specific brain regions or electrode clusters in each task, and whether consistent or task-dependent patterns emerge across datasets? The paper provides such analysis only for two datasets in the appendix, whereas this should be a central focus of the experimental discussion.

Q2. The introduction mentions spatial relationships, functional connectivity, and disease localization as motivations, yet the concrete application scenarios remain vague. In what practical contexts does spatial preservation provide the most tangible benefits? Given the lack of a clear link between the spatial learning mechanism and downstream functional or clinical outcomes, the motivation currently appears more algorithm-driven than application-driven.

Q3. What do the authors see as the greatest potential real-world value of spatially preserving EEG representations?

---

> ### Author Response · Authors · 2025-11-21
> **Rebuttal (1/2)**
>
> Thank you very much for your thoughtful review. This work sits at the intersection of deep learning and EEG, and our focus has been primarily on the deep learning aspects. However, the EEG perspective is equally crucial, and the points raised in your review have been very valuable for broadening our view. We have updated our submission based on reviews.
>
> **W1 and W2: Limitation of coherence-based pseudo-labels and temporal modeling.**
>
> We agree that coherence as a target has inherent limitations, and we are aware of more sophisticated measures such as partial directed coherence (Baccalá \& Sameshima, 2001). At the same time, our main goal in this paper is to introduce a framework that is substantially different from prior SSL approaches for EEG, and incorporating additional complex notions such as partial directed coherence and Granger causality in this single work risks diluting the core conceptual contribution of the framework. Conceptually, the effectiveness of a fixed coherence target is related to studies in the field of masked modeling, where masking induces variability and randomness that function as augmentations of the original samples. Nevertheless, partial directed coherence and Granger causality are indeed important directions, and we agree that they might lead to interesting future work when combined with our study.
>
> To shed more light on temporal–frequency dynamics in our model, we have added comparisons in Table 5 using temporal correlation and phase synchronization as alternative targets. Empirically, we observe that both phase synchronization and temporal correlation degrade the baseline performance (by **0.7** and **0.2**, respectively), which is consistent with previous work showing that temporal–frequency pairs are particularly helpful for learning robust features (Zhang et al., 2022).
>
> **W3: Band-specific physiological characteristics.**
>
> Indeed, different frequency bands carry important cognitive and physiological information, which is especially relevant for supervised learning. In our case, band selection and band-mixture design are applied only during pre-training, where the task is to predict spatial relationships. For downstream medical label prediction, the model is still trained (via linear probing or fine-tuning) using all frequency bands, so no band mixture constraints are imposed in the downstream tasks.
>
> **W4: Generalizability.**
>
> To assess the generalizability of our framework, we specifically compare multiple decoders to show that the performance gains of our SSL model are not tied to the particular capacity or design of the downstream architecture. We run additional experiments with a decoder using global average pooling and another using learned weighted pooling on the STEW and CHB-MIT datasets, and compare these three commonly used decoder types in the table:
>
> | Method             | STEW Accuracy | STEW AUROC | CHB-MIT B-Accuracy | CHB-MIT AUROC |
> |--------------------|--------------|------------|---------------------|---------------|
> | Attention (Default) | 75.3(0.9)    | 83.0(0.8)  | 54.7(1.2)          | 76.3(4.1)     |
> | +SPR               | 78.3(1.9)    | 86.3(2.5)  | 89.6(0.8)          | 94.1(1.4)     |
> | Global average     | 74.3(0.4)    | 82.2(0.5)  | 53.9(1.5)          | 76.6(2.7)     |
> | +SPR               | 78.7(0.7)    | 86.6(0.4)  | 88.8(1)            | 95.1(1.1)     |
> | Learned weighted   | 72.4(1)      | 80.1(1.1)  | 55.6(1.9)          | 74.8(2.7)     |
> | +SPR               | 77.8(1.5)    | 85.7(1.5)  | 87.6(0.9)          | 92.9(1)       |
>
>
> For each decoder, we first report the purely supervised performance and then the performance when combined with our SPR framework. The resulting accuracy gains are 3\%, 4.4\%, and 5.4\% on STEW, and 34.9\%, 34.9\%, and 32\% on CHB-MIT, respectively. These results show that our method consistently improves all supervised baselines by a substantial margin.

---

> ### Author Response · Authors · 2025-11-21
> **Rebuttal (2/2)**
>
> **Q1: Brain regions and visualization.**
>
> It is challenging for us to make very general neuroanatomical statements without a specific scenario. Here, we add an explanation of the CHB-MIT results. We examine the neuroanatomy more closely using visualizations (Fig. 7D and Fig. 8 in Appendix B) and observe structured activation patterns highlighting functionally connected brain regions that are relevant for identifying focal epilepsy (Watanabe, 1989; Plummer et al., 2008; Foldvary et al., 2001). In Fig. 8, we see more pronounced local activation over the **frontal** and  **prefrontal** areas, as well as the  **temporal** lobe in the $\beta$ and $\gamma$ bands, and these regions show correspondingly elevated local connectivity in Fig. 7D. By contrast, central and parietal regions exhibit relatively weak activation and limited local connectivity in the learned heat map. Overall, the learned heat map reveals dominant local connectivity rather than long-range interactions, which is consistent with our design choice to emphasize local connectivity through higher-frequency coherence.
>
> **Q2 and Q3: Application scenarios and real-world value.**
>
> Building on the answer for Q1, datasets containing focal pathology or conditions with localized abnormalities are natural application scenarios for our method, because the design explicitly focuses on representing local spatial relationships rather than long-range patterns. It is also reasonable to expect even greater value when dealing with signals of higher spatial resolution, which is an area that has received relatively little attention in the current SSL EEG literature and could become an important future direction.
>
> Another concrete example is when researchers restrict their analysis to frequencies up to, for instance, 40 Hz due to noise constraints, sampling rate, filtering choices, or a focus on cognitive paradigms. In such settings, there is a risk of discarding spatially specific information residing in the higher
> $\gamma$ band, whereas our model is designed to capture and preserve these spatial features. Consequently, the features produced by our model can be used as auxiliary inputs, even in non–machine learning pipelines, to reintroduce spatial information that would otherwise be omitted. This illustrates both the broad applicability of our framework and the motivation for learning a spatiality-preserving representation in the first place. We have provided our code in the supplementary materials and will release both code and pretrained model weights upon publication. We have made a strong effort to ensure reproducibility and are committed to further supporting the EEG community in using our code and pretrained models. We sincerely hope that this work will help pave the way for leveraging powerful deep learning models to supply spatial features that assist EEG researchers beyond the deep learning community.

---

> > ### Author Response · Authors · 2025-11-25
> > **References**
> >
> > [1] Baccalá, L. A., & Sameshima, K. (2001). Partial directed coherence: a new concept in neural structure determination. Biological cybernetics, 84(6), 463-474.
> >
> > [2] Zhang, X., Zhao, Z., Tsiligkaridis, T., & Zitnik, M. (2022). Self-supervised contrastive pre-training for time series via time-frequency consistency. Advances in neural information processing systems, 35, 3988-4003.
> >
> > [3] Watanabe, K. (1989). The Localization‐Related Epilepsies: Some Problems with Subclassification. Psychiatry and Clinical Neurosciences, 43(3), 471-475.
> >
> > [4] Plummer, C., Harvey, A. S., & Cook, M. (2008). EEG source localization in focal epilepsy: where are we now?. Epilepsia, 49(2), 201-218.
> >
> > [5] Foldvary, N., Klem, G., Hammel, J., Bingaman, W., Najm, I., & Luders, H. (2001). The localizing value of ictal EEG in focal epilepsy. Neurology, 57(11), 2022-2028.

---

> > > ### Comment · Reviewer_4ps2 · 2025-11-25
> > >
> > > I thank the authors for their detailed rebuttal and the additional experiments. While I appreciate the effort to demonstrate architectural robustness, I feel that the core concern regarding generalizability (W4) has not been fully resolved. Comparing different decoders does not fundamentally address the need for validation across diverse neurophysiological paradigms.
> > >
> > > Furthermore, the authors' admission that their focus is "primarily on deep learning aspects" rather than EEG interpretation reinforces my concern that the physiological validity of the method might be limited. The work appears to be an algorithm-first engineering application that overlooks the complexity of brain signals and neurophysiological paradigms.
> > > Therefore, I will maintain my original score.

---

### Meta-Review · Area_Chair_UPXG · 2026-01-04

**Summary:**

This paper introduces Spatiality Preserving Representation (SPR) Learning, a self-supervised learning framework for EEG that aim to address the loss of spatial information in existing methods. The key idea is a coherence pseudo-label prediction task where the model learns to predict band-limited (alpha, beta, gamma) magnitude-squared coherence between electrode pairs. In other words, targets are obtained with a softmax applied to cross-spectrum matrices. The authors report improvements of 4.7%, 9.7%, 1.6%, and 15.6% over state-of-the-art methods across four EEG datasets (STEW, TUAB, TUEV, CHB-MIT).

The four reviewers had divergent opinions. Reviewers uzH8 and jRWD (both Score: 6) found the coherence pseudo-label approach novel and well-motivated, with strong empirical results. Reviewer 4ps2 (Score: 4) expressed concerns about the algorithm-first approach and limited application breadth. Reviewer ePRy (Score: 2) raised fundamental concerns about experimental validation, questioning whether the architecture or the SSL task drives performance gains, and requested control experiments comparing the encoder with standard SSL methods.

**Rationale for decision:** While this paper presents a novel contribution to self-supervised learning for EEG by introducing spatial coherence prediction as a pretext task, the overall reviewer consensus does not support acceptance. The key factors informing this decision are:

**1. Split Reviewer Opinion with Unresolved Concerns:** Only two of four reviewers (uzH8, jRWD) were supportive with scores of 6. Reviewer 4ps2 explicitly maintained their score of 4 after the rebuttal, and Reviewer ePRy (initial score 2) remained skeptical despite acknowledging the SSL method improves performance, stating it is "in and of itself, not yet useful for the community."

**2. Evaluation Methodology Concerns Not Fully Resolved:** Reviewer ePRy raised fundamental questions about the validity of the SOTA claims that were not adequately addressed:
- The comparison mixes in-domain and cross-domain methods inconsistently, making it difficult to interpret results
- LaBraM and CBraMod are missing from STEW and TUEV evaluations, yet the paper claims SOTA performance
- The explanation for why prior baselines underperform due to "channel mismatches" remains unclear in the in-domain setting where train and test use identical channel layouts

**3. Limited Application Scope:** Reviewer 4ps2 correctly noted that the four datasets share similar characteristics (low-density, short-segment EEG with comparable sampling rates and channel counts). The lack of validation on diverse paradigms such as motor imagery, SSVEP, or neurodegenerative disease datasets limits confidence in the claimed generalizability.

**4. Algorithm-First vs. Neurophysiological Grounding:** Both Reviewers 4ps2 and ePRy expressed concerns that the work prioritizes algorithmic novelty over demonstrated practical utility and neurophysiological validity. The authors acknowledged their focus is "primarily on deep learning aspects," which may not meet the expectations for work in the neuroscience applications track.

**5. Incremental Gains on Some Datasets:** While the 15.6% improvement on CHB-MIT is impressive, gains on other datasets are more modest (1.6-4.7%). The band-mixture ablation shows only marginal benefit on STEW (0.3%), suggesting the design choices may not consistently provide advantages across different settings.

**Constructive Feedback for Revision:**
- Provide complete and consistent comparisons with all relevant baselines (including LaBraM, CBraMod) across all datasets
- Clarify the in-domain vs. cross-domain experimental setup more explicitly
- Validate on additional paradigms (MI, SSVEP) to demonstrate broader applicability
- Strengthen the connection between the learned representations and neurophysiological interpretability

The coherence pseudo-label approach is a promising direction that deserves further development, but the current evaluation does not sufficiently establish the method's practical relevance and superiority over existing approaches.

**Reviewer Concerns:**

### Addressed by Rebuttal:
- **Control Experiments with Standard SSL** (ePRy): Authors conducted the critical control experiment—using their encoder with RP (Banville et al.), SimCLR, and VICReg instead of their coherence task. Results show all traditional SSL methods underperform the supervised baseline (best: SimCLR 74.9% vs. supervised 75.3%), while SPR achieves 78.3%. This suggests the SSL task, not just the architecture, drives the gains.
- **Decoder Generalization** (jRWD, 4ps2): Authors tested three decoder architectures (attention pooling, global average pooling, learned weighted pooling). SPR consistently improves all supervised baselines: +3-5.4% on STEW, +32-35% on CHB-MIT, showing the SSL benefit is decoder-agnostic.
- **Alternative Connectivity Measures** (uzH8, jRWD): Authors compared coherence against temporal correlation and phase synchronization as prediction targets. Both alternatives degraded performance (by 0.2 and 0.7 respectively), supporting the choice of coherence.
- **Band Selection Ablation** (uzH8, ePRy): Authors provided ablation on STEW and CHB-MIT. Using full-band (all Hz) instead of alpha-beta-gamma mixture reduces CHB-MIT performance by 3.5%, validating the band-mixture design for capturing spatially-specific high-frequency information.
- **Cross-Domain Pretraining** (jRWD): Authors added cross-domain experiments (TUAB + TUEV), showing B-accuracy increases and variance decreases with more pretraining samples, demonstrating scalability of the coherence pretext task.
- **Spatial Pattern Interpretability** (uzH8, jRWD): Authors provided some analysis of activation maps (Fig. 7D, Fig. 8) showing structured patterns over functionally connected regions relevant to focal epilepsy, with predominantly local rather than long-range connectivity consistent with the design focus on higher-frequency coherence.
- **Volume Conduction** (uzH8): Authors clarified that volume conduction is addressed through bipolar montage referencing in preprocessing, following clinical recommendations.
- **SPR(Random) Performance** (ePRy): Authors explained that their supervised model outperforms some prior SSL methods on certain datasets due to channel-order insensitivity of their architecture, but importantly, SPR still provides consistent 3-5% gains over this supervised baseline.

**Reviewer Scores:**

| Reviewer | Initial Score | Predicted Post-Discussion Score |
|----------|---------------|--------------------------------|
| **4ps2** | 4 (Marginally Below) | **4** - Reviewer explicitly maintained score after rebuttal, citing concerns about application breadth and algorithm-first approach. Stated "I will maintain my original score." |
| **ePRy** | 2 (Reject) | **3-4** - Despite authors' control experiments, reviewer's final substantive comment stated: "it is clear that your SSL pretraining method improves downstream performance. This is interesting, but in and of itself, not yet useful for the community." Reviewer maintained concerns about evaluation scheme validity, in-domain vs. cross-domain comparison confusion, and incomplete SOTA comparisons (missing LaBraM/CBraMod on STEW/TUEV). Likely remains skeptical. |
| **uzH8** | 6 (Marginally Above) | **6-7** - Initial review was positive, finding coherence pseudo-label "conceptually interesting." All concerns (band selection, volume conduction, interpretability) were addressed in rebuttal. Reviewer was muted before confirming satisfaction but had no fundamental objections. |
| **jRWD** | 6 (Marginally Above) | **6-7** - Initial review found the method novel with "promising performance improvements." All questions (decoder analysis, interpretability, temporal correlation comparison) were addressed comprehensively. Reviewer was muted before responding. |

---

### Decision · Program_Chairs · 2026-01-26

Reject